

# The Zugspitze radiative closure experiment for quantifying water vapor absorption over the terrestrial and solar infrared.
# Part I: Setup, uncertainty analysis, and assessment of far-infrared water vapor continuum

Ralf Sussmann, Andreas Reichert, and Markus Rettinger

Karlsruhe Institute of Technology, IMK-IFU, Garmisch-Partenkirchen, Germany

*Correspondence to*: Ralf Sussmann (ralf.sussmann@kit.edu)

**Abstract.** Quantitative knowledge of water vapor radiative processes in the atmosphere throughout the terrestrial and solar infrared spectrum is still incomplete even though this is crucial input to the radiation codes forming the core of both remote sensing methods and climate simulations. Beside laboratory spectroscopy, ground-based remote sensing field studies in terms of so-called radiative closure experiments are a powerful approach, because this is the only way to quantify water absorption under cold atmospheric conditions. For this purpose, we have set up at Mt. Zugspitze (47.42 °N, 10.98 °E, 2964 m a.s.l.) a long-term radiative closure experiment designed to cover the infrared spectrum between 400 to 7800 $cm^{-1}$ (1.28-25 µm). As a benefit for such experiments, the atmospheric states at Zugspitze frequently comprise very low integrated water vapor (IWV; minimum = 0.1 mm, median = 2.3 mm) and very low aerosol optical depth (AOD = 0.0024-0.0032 at 7800 $cm^{-1}$ at airmass 1). All instruments for radiance measurements and atmospheric state measurements are described along with their measurement uncertainties. Based on all parameter uncertainties and the corresponding radiance Jacobians, a systematic residual radiance uncertainty budget has been set up to characterize the sensitivity of the radiative closure over the whole infrared spectral range. The dominant uncertainty contribution in the spectral windows used for far-infrared (FIR) continuum quantification is from IWV uncertainties, while *T*-profile uncertainties dominate in the mid-infrared (MIR). Uncertainty contributions to near-infrared (NIR) radiance residuals are dominated by water vapor line parameters in the vicinity of the strong water vapor bands. The window regions in between these bands are dominated by solar FTIR calibration uncertainties at low NIR wavenumbers, while uncertainties due to AOD become an increasing and dominant contribution towards higher NIR wavenumbers. Exceptions are methane or nitrous oxide bands in the NIR, where the associated line parameter uncertainties dominate the overall uncertainty.

As a first demonstration of the Zugspitze closure experiment, a water vapor continuum quantification in the FIR spectral region (400–580 $cm^{-1}$) has been performed. The resulting FIR foreign continuum coefficients are consistent with the MT_CKD 2.5.2 continuum model and also agree with the most recent atmospheric closure study carried out in Antarctica. Results from the first determination of the NIR water vapor continuum in a field experiment are detailed in a companion



paper (Part III) while a novel NIR calibration scheme for the underlying FTIR measurements of incoming solar radiance is presented in another companion paper (Part II).

## 1 Introduction

Water vapor causes about 60 % of the telluric greenhouse effect and about 72 % of the atmospheric absorption of incoming
solar radiation for clear skies (Kiehl and Trenberth, 1997). Furthermore, water vapor feedback approximately doubles the response of surface temperature to the imposition of an external forcing, e.g. anthropogenic $CO_2$ emissions (Held and Soden, 2000). Finally, water vapor is a target species for spectrometric remote sensing methods based on the differential optical absorption principle, and, due to its high variability, it is also a potential interfering species for remote sensing of all other atmospheric trace gases (Frankenberg et al., 2008; Sussmann and Borsdorff, 2007; Sussmann et al., 2011). All in all,
it is important to put efforts toward a quantitative understanding of all details of water vapor absorption throughout the whole terrestrial and solar infrared spectrum.

Numerical approaches dedicated to weather forecast, climate prediction, and remote sensing data analysis are based upon radiative transfer codes calculating the absorption and/or emission of radiation by atmospheric trace gases, aerosols, and clouds as a function of wavelength. Modeling the radiative impact of the gas phase molecular compounds has to include
two radiative processes: pure rotational absorption/emission in the far infrared (FIR) and vibration-rotation absorption/emission in the mid-infrared (MIR) and the near infrared (NIR). According to quantum-mechanical selection rules, both processes lead to atmospheric band-type absorption/emission spectra with thousands of individual spectral lines. The most accurate (but time-consuming) way of simulating these processes is a fully resolved line-by-line approach, e.g. via the widely used Line-By-Line Radiative Transfer Model LBLRTM (Clough et al., 2005; Mlawer et al., 2012). The
LBLRTM is then used as validation reference for the faster Rapid Radiative Transfer Model (RRTM) which avoids time–consuming line-by-line calculations by a correlated-k approach (Mlawer et al., 1997) and is used within many climate models (i.e. general circulation models).

However, there are still uncertainties which are potentially aliased into the applications (climate simulations, weather forecast, remote sensing). One class of uncertainties is related to the spectroscopic line parameters (e.g. line strength and
pressure broadened half width). E.g. the current spectroscopic foundation of LBLRTM is the line parameters database aer_v_3.2 which is built from HITRAN 2008 (Rothman et al., 2009) with notable exceptions for $H_2O$, $CO_2$, $CH_4$, and $O_2$ (for details see http://rtweb.aer.com/line_param_whats_new.html). Another source of uncertainty is the so-called continuum absorption, especially due to water vapor. It is a spectrally less structured contribution dominating in window regions (e.g. Shine et al., 2012) which comprises two components, the self continuum (attributed to $H_2O$-$H_2O$ interactions)
and the foreign continuum (attributed mainly to $H_2O$-$N_2$ interactions). Although still no definite continuum theory does exist, it seems that a consensus has been reached on the existence of two physical processes contributing, namely i)





monomer contributions resulting from perturbations of the line shape due to (self and foreign) pair-interactions during molecular collisions and ii) dimer contributions, i.e. absorption due to stable and/or metastable water dimers. Evidence for existence of water dimers in the atmosphere has been reported by Pfeilsticker et al. (2003) and Ptashnik (2008). However, the relative importance of the monomer and dimer contributions as a function of temperature and wavenumber (especially

for window versus in-band regions) are far from being understood. For recent reviews see Shine et al. (2012), Mlawer et al. (2012), and references therein. The most widely used water vapor continuum model at this time (MT_CKD 2.5.2) is based on the monomer hypothesis, while contributions from water dimers shall be implemented in future versions (Mlawer et al., 2012). MT_CKD is a semiempirical model combining a collision-induced component and a line shape component. In both terms empirical parameters are set in a way to achieve agreement with laboratory and field measurements. Constraining

measurements have hitherto been restricted to measurements within the microwave, the FIR, the MIR and, recently, also the NIR, see Mlawer et al. (2012) for details and references. This means that considerable fractions of the full 0-20000 cm$^{-1}$ range of MT_CKD are semiempirical extrapolations in between the constraining measurements, i.e. continuum parameters reported there are uncertain.

The potential impact of line parameter or continuum model uncertainties has been investigated in a series of papers. E.g.

one study investigated the impact of improved NIR water vapor line parameters in simulations with the ECHAM4 general circulation model (Lohmann and Bennartz, 2002). They found that the global annual mean atmospheric absorption of solar radiation in ECHAM4 is increased under all skies between 3.2 and 3.7 W m$^{-2}$ and between 5.0 and 5.7 W m$^{-2}$ under clear skies for the different data sets. While the dynamics barely change, the hydrological cycle is slightly weaker, the cloud cover has decreased by 0.4 % and the precipitation by 0.06 to 0.08 mm d$^{-1}$ with the new data set. A FIR continuum study

showed that modifications to the previously derived strength of the water vapor continuum in the 10–700 cm$^{-1}$ region within the Community Earth System Model (CESM) had a statistically significant impact on both the radiation and dynamics with changes in the vertical structure of temperature, humidity, and cloud amount, all of which impacted the diabatic heating profile (Turner et al., 2012a). Paynter and Ramaswamy (2012) showed that the water vapor continuum could result in between 1.1 W m$^{-2}$ and 3.2 W m$^{-2}$ additional clear-sky absorption of solar radiation globally. According to

Paynter and Ramaswamy (2014), this sizable range is due to fairly large measurement uncertainties in the shortwave near-infrared window regions (Ptashnik et al., 2004, 2011, 2012, 2013; Paynter et al., 2007, 2009; Baranov and Lafferty, 2011, 2012; Mondelain et al., 2013). After inclusion of a modified parameterization for the shortwave water vapor continuum (BPS-MTCKD 2.0) to the Geophysical Fluid Dynamics Laboratory (GFDL) global model, Paynter and Ramaswamy (2014) found the surface energy budget adjusted predominantly through a decrease in both surface latent and sensible heat. This

leads to a decrease in tropical convection and a subsequent 1 % reduction in tropical rainfall. Finally, a recent NIR continuum study investigated the impact of switching from the CKD continuum model frequently used in climate models to a continuum model where absorption is enhanced at wavelengths greater than 1 μm based on recent measurements (CAVIAR). They found that for CKD and CAVIAR respectively, and relative to the no-continuum case, the solar



component of the water vapor feedback is enhanced by about 4 and 9 %, the change in clear-sky downward surface irradiance is 7 and 18 % more negative, and the global-mean precipitation response decreases by 1 and 4 % (Rädel et al., 2015).

Due to the critical relevance of line parameter and continuum model uncertainties for climate simulations a series of quality measure experiments has been performed. Such field closure studies comprise high-spectral-resolution radiance measurements and radiative transfer simulations of the measured spectra driven by coincident atmospheric state measurements of integrated water vapor (IWV) and other relevant parameters. As part of the U.S. Atmospheric Radiation Measurement (ARM) program (Ackermann and Stokes, 2003) a series of radiative closure experiments has been setup (e.g. Turner et al., 2004; 2012b) which was complemented by the Italian ECOWAR (Earth COoling by WAter vapor Radiation) project (e.g. Bhawar et al., 2008; Bianchini et al., 2011). Various experiments have addressed the quality of (water vapor) line parameters in the FIR (Esposito et al., 2007; Delamere et al., 2010; Masiello et al., 2012), the water vapor continuum in the FIR (Tobin et al., 1999; Serio et al., 2008; Delamere et al., 2010; Liuzzi et al., 2014), and the water vapor continuum in the MIR (Turner et al., 2004; Rowe et al., 2006; Rowe and Walden, 2009). A crucial requirement for radiative closure experiments in the FIR and MIR is to select a site guaranteeing a wide range of IWV levels including the occurrence of very low IWV levels. Dry atmospheric states (IWV < 1 mm) are highly beneficial to attain information on absorption coefficients in otherwise saturated spectral regions (e.g. the pure rational water band of water vapor). For these reasons, there have been dedicated campaigns performed in dry regions on the globe, e.g. at the Sheba ice station (Tobin et al., 1999) or the RHUBC II campaign in the Atacama desert (Turner and Mlawer, 2010).

Coming to the NIR we note that for this spectral region to our knowledge no atmospheric radiative closure experiments have been reported in the literature with the exception of one study by Sierk et al. (2004). A hindrance for quantitative field studies may have been the fact that absorption in the NIR due to aerosols can become comparable to the magnitude of the water vapor continuum absorption of interest (Ptashnik et al., 2015). The possibility to accurately separate these two components depends on aerosol load (i.e. aerosol optical depth, AOD) and therefore on field site characteristics. We will come back to this when introducing the new Zugspitze field experiment (including NIR measurements) below. On the other side, there have been many laboratory studies in the NIR range. Laboratory experiments using FTIR spectrometry and large cells have shown that the self-continuum within bands contains more spectral structure than given by the MT_CKD model (Ptashnik et al., 2004; Paynter et al., 2009). The self-continuum absorption within the windows was found to be an order of magnitude stronger than given by MT_CKD and to vary from window to window significantly less. (Baranov and Lafferty, 2011; Ptashnik et al., 2011, 2013). Also the NIR foreign continuum in window regions was found to be about an order of magnitude stronger compared to MT_CKD (Ptashnik et al., 2012). Another issue is the obvious discrepancy between laboratory measurements performed by different techniques. E.g. the magnitude of the self continuum in the 1.6 and 2.1 µm windows derived from laboratory FTIR spectrometry (Baranov and Lafferty, 2011; Ptashnik et al., 2011) is higher by about one order or magnitude as compared to results obtained by cavity ring-down spectroscopy (CRDS; Mondelain et al., 2013;





2015). Furthermore, CDRS results differ to laboratory results obtained by calorimetric interferometry (Bicknell et al., 2006) by a factor of 4–5. Reasons behind these inconsistent laboratory results could be the differing physical measurement principles and/or were tentatively attributed to surface effects (e.g. water on walls, water droplets or clusters in the cell; Ptashnik et al., 2013; 2015). Finally, a drawback common to all laboratory measurements is that they have to be performed

at least at room temperature or even be heated, in order to detect the weak continuum absorption in the limited optical path length of the cells. Therefore, for climate and remote sensing applications an extrapolation of continuum coefficients to the lower atmospheric temperatures is required. This, however, is a problem because the observational evidence of a negative exponential temperature dependency of the self continuum still cannot be described by a physical model in a quantitative manner (e.g. Shine et al., 2012).

Our review of previous activities to mature the quantitative knowledge of water vapor absorption indicates a need for further radiative closure studies in order to i) validate/complement the previous studies in the FIR and MIR and ii) establish a NIR closure experiment in the field in order to provide an independent assessment of the existing but differing laboratory results with respect to their mutual agreement and the agreement versus MT_CKD under atmospheric conditions.

The goal of this paper is therefore to report on a new water vapor radiative closure experiment set up on the summit of Mt.
Zugspitze (47.42 °N, 10.98 °E, 2964 m a.s.l.) covering the FIR, MIR and NIR spectral range. This experiment is not a campaign but designed as a long-term (multi-annual) study with the benefit to attain improved data statistics compared to campaigns. Furthermore, the Zugspitze is a unique site as it is not remote (accessible by cable car) but offers at the same time extraordinarily dry conditions and low aerosol loads. As outlined before, these are crucial prerequisites for closure studies and on dry winter days the Zugspitze offers regularly conditions comparable to the driest sites and sites with the
highest atmospheric transparency on the globe. The heritage of the Zugspitze site and team is linked to ground-based solar FTIR remote sensing with some focus on water vapor (e.g. Sussmann et al., 2009; Vogelmann et al., 2011; Vogelmann et al., 2015). The Zugspitze solar FTIR is part of the Network of the Detection of Atmospheric Composition Change (NDACC; www.ndacc.org) which also comprises a working group on water vapor sounding techniques (e.g. Kämpfer, 2013). This paper describes an extension of the Zugspitze instrumentation including the NDACC solar FTIR system
(Sussmann and Schäfer, 1997) adapted for NIR radiance measurements and complemented by additional instruments for FIR and MIR radiance measurements and IWV sounding as well as further measurements of the atmospheric state.

Our publication on the Zugspitze radiative closure experiment comprises a set of 3 companion papers thereafter designated as Part I, II, and III, respectively. This paper (Part I) illustrates the basic idea and setup within Sect. 2. Section 3 details the radiance measurements in the FIR, MIR, and NIR, followed by Sect. 4 describing the state measurements, Sect. 5 the
radiative transfer calculations, and Sec. 6 giving a detailed uncertainty analysis. Finally, Sect. 7 shows an example closure study in the FIR, and the results are compared to MT_CKD and other field measurements. Part II is on a novel calibration




scheme for solar FTIR radiance measurements, and Part III gives the application of this to a NIR closure study, with the results on the NIR water vapor continuum compared to MT_CKD and laboratory measurements.

**2 Idea and setup of the closure experiment**

At the summit of Mt. Zugspitze (47.42 °N, 10.98 °E, 2964 m a.s.l.) we have set up spectral radiance measurements covering the FIR, the MIR, and the NIR along with atmospheric state measurements, most importantly IWV (Fig. 1 and Tab. 1). The idea of the closure experiment is to compare measured radiance spectra with simulations of the spectra driven by coincident state measurements. Minimization of measured minus simulated spectral radiance (thereafter referred to as "spectral residuals") leads to improved water vapor absorption parameters used in the radiance simulations (Fig. 2). The basic principle behind this approach has been presented before (e.g. Tobin et al., 1999; Turner and Mlawer, 2010). However, there are 4 aspects which are special to our Zugspitze setup:

i) Very dry atmospheric conditions are a pre-requisite for closure studies of this kind due to the otherwise saturated spectral regions (see e.g. Fig. 1 in Tobin et al., 1999). To achieve this goal previous campaigns were performed at remote locations like the Sheba ice station (Tobin et al., 1999) or at the Atacama desert where IWV levels down to 0.2 mm were achieved (Turner and Mlawer, 2010). On the other hand, at the Zugspitze we frequently encounter comparably dry atmospheric conditions (min IWV = 0.1 mm, see Fig. 3 and Tab. 2), but the Zugspitze is at the same time an easy-to-access site, which can be reached within 20 min by cable car from our institute´s building at Garmisch-Partenkirchen. Note that the minimum IWV levels at Zugspitze (0.1 mm) are approximately a factor of 40 lower than at typical lowland mid-latitude sites.

ii) Unlike previous campaign-type studies, our field experiment is designed as a long-term study (time scale ~10 years) – this is beneficial for attaining improved measurement statistics.

iii) The Zugspitze radiative closure experiment is – to our knowledge for the first time – extended to include the NIR spectral range, while previous studies had focused on the MIR (e.g. Tobin et al., 1999) and FIR (e.g. Delamere et al., 2010), respectively.

iv) A benefit of the Zugspitze high-altitude mountain site is that AOD is typically very low, i.e. about a factor of 10 lower than at typical lowland mid-latitude sites. This is important because otherwise in the NIR the AOD would become significantly higher than the water vapor continuum optical depth and this would be a hindrance for accurate continuum quantification in the NIR (Ptashnik et al., 2015). The AOD levels encountered in the Zugspitze closure data set used in this study (i.e. dry clear sky days within the time span Dec 2013–Feb 2014, see Sect. 7.1 for data selection details) are in the range 0.0005–0.00075 at 2500 cm$^{-1}$ and in the range 0.0024-0.0032 at 7800 cm$^{-1}$ at airmass 1.



## 3 Spectral radiance measurements

### 3.1 FIR and MIR radiance measurements

Downwelling thermal emission is measured in the FIR and MIR spectral range from 400 to 3000 cm$^{-1}$ (25–3.3 µm) via an Extended-range Atmospheric Emitted Radiance Interferometer (E-AERI). This instrument was designed by the University of Wisconsin Space Science and Engineering Centre and is manufactured by ABB Bomem Inc. (Quebec, Canada). Details of the instrument design and performance have been given by Knuteson et al. (2004a; 2004b). Such instruments have been operated, e.g. at the SHEBA Ice Station (Tobin et al., 1999), in the Atacama Desert (Turner and Mlawer, 2010), or at Eureka (Mariani et al., 2012). Briefly, the instrument inside the Zugspitze container is based on a 0.5-cm$^{-1}$-resolution [maximum optical path difference (OPD$_{max}$) of 1 cm] FTIR spectrometer. The interferometer front window is linked to the frontend which is mounted outside the container in the so-called through-wall configuration and comprises the scene mirror and two blackbodies (BB), at ambient temperature and at 310 K, respectively (Fig. 2). The frontend hatch used to protect the scene mirror against precipitation has been modified from its original flat-roof shape to a pitched-roof shape in order to avoid snow accumulations. Scan duration for one interferogram is 2 sec and the total repeat cycle is 10 min, with 4 min integration for the atmospheric observations, and 2 times 2 min for the blackbody measurements.

Radiometric calibration of the E-AERI is performed via the approach by Revercomb et al. (1988). The related FIR and MIR radiometric uncertainty specifications are given in Tab. 1 and more details will be presented in Sect. 6.1. Briefly, there is a known radiometric bias in the E-AERI radiance measurements which was corrected via the method proposed by Delamere et al. (2010). This method relies on the assumption that a fraction $f$ of the instrument's field of view is obstructed by instrument parts. The value of $f$ is constrained by a fit to measured radiance in the 827 to 835 cm$^{-1}$ spectral window. We obtain $f = 0.0049$ which is then used for performing the bias correction according to Delamere et al. (2010).

An estimate of radiance measurement noise of the E-AERI is obtained as an output from the calibration procedure, see Sect. 6.1 for numbers. The spectral radiance noise can be reduced using a filter based on principal component analysis as outlined in Antonelli et al. (2004) and Turner et al. (2006). Based on 8000 Zugspitze spectra, this analysis indicated that the use of the first 239 principal components is optimal. This resulted in a ~50% noise reduction.

### 3.2 NIR radiance measurements

Solar absorption spectra in the NIR spectral range from 2500 – 7800 cm$^{-1}$ (4.0-1.28 µm) were implemented via the Zugspitze high-resolution solar FTIR system based on a Bruker IFS 125 HR interferometer with an optical path difference up to 418 cm (Sussmann and Schäfer, 1997). This instrument is operational since 1995 for spectrometric MIR trace gas measurements within the NDACC network. All details of the new NIR radiometric measurements are given in Part II. Briefly, the NIR operations are utilizing an InSb detector along with a KBr beamsplitter (InGas/CaF$_2$ optional), interferograms are recorded with a OPD$_{max}$ of 45 cm and averaged over 4 to 8 scans for one spectrum (75–150 seconds



integration time). Radiometric calibration is achieved by a novel approach utilizing a combination of the Langley calibration method and a hot blackbody calibration source (< 2000 K) used for interpolating the calibration curve between the individual spectral Langley calibration points (see Part II for detailed information). Related NIR radiometric uncertainties are given in Tab. 1 and will be further discussed in Sect. 6.1.

## 4 State measurements

### 4.1 Integrated water vapor and water vapor profiles

For the closure experiments based on E-AERI radiance measurements in the FIR and MIR, IWV is directly retrieved from E-AERI spectra. This allows for an ideal spatiotemporal matching between the radiance measurements in the terrestrial infrared and the corresponding IWV state measurements. IWV is retrieved by minimizing E-AERI vs. LBLRTM spectral residuals in IWV-sensitive windows. For this purpose we implemented an approach similar to the method proposed by Serio et al. (2008). Details of the IWV retrieval and the procedure for selection of suitable spectral windows are outlined in Appendix A. Numbers for the uncertainty of the E-AERI-based IWV retrieval are given in Tab. 1. The underlying uncertainty analysis is given in Appendix A, and Sect. 6 derives the related radiance uncertainty.

For the NIR closure measurements (Part III), IWV was retrieved directly from the solar FTIR spectral radiance measurements (see Sect. 3.2) using a MIR retrieval scheme which exploits several spectral micro-windows in the 2610 – 3050 cm-1 range (Schneider et al. 2012; 2016). Again, this allows for an ideal spatiotemporal matching of the solar infrared radiance measurements and the correlative IWV state measurements. Specifications of the uncertainty of the IWV retrieval from the solar FTIR are given in Tab. 1 and in Sect. 6.3, where also the related radiance uncertainty is presented.

Profile shape information on water vapor was taken from four-times-daily National Center for Environmental Prediction (NCEP) resimulation data. The reason for not using water vapor profiles from the LHATPRO microwave radiometer (Radiometer Physics, Germany; Rose et al., 2005) available on site is that a comparison of LHATPRO water vapor profiles with coincident NCEP resimulation profiles for the FIR continuum data set resulted in relatively large discrepancies, i.e. a mean precision (2-$\sigma$) of 27.6 % and a mean bias of 20.4 %. We therefore use NCEP profiles throughout the closure study. However, a comparison with LHATPRO profiles is used in order to detect and discard atmospheric states in which NCEP fails to realistically cover spatiotemporal variability of water vapor (see Sect. 7.1). An estimate of the NCEP profile shape uncertainty based on a comparison with radiosonde profiles is given in Tab. 1 and derived in Sect. 6.3.

### 4.2 Temperature profiles

Temperature profiles for the radiative transfer calculations were based on four-times-daily pressure-temperature-humidity profiles from NCEP interpolated to the time of the radiance measurement. Since the lowest atmospheric layer above the Zugspitze summit is certainly influenced by the mountain surface, deviations between the true temperature profile and



NCEP are expected. In order to account for this effect, the NCEP profile was corrected for the lowermost 500 m above the Zugspitze summit. The correction is retrieved using the spectral radiance observed by the E-AERI in the central part of the 15-μm band of $CO_2$ (i.e. $625 - 715$ cm$^{-1}$). Because of the strong absorption, the measured radiance in this spectral region strongly correlates to the temperature of the environment close to the instrument. We use for this kind of boundary layer

temperature inversion the retrieval scheme developed by Esposito et al. (2007), which has successfully been utilized by a series of studies (Serio et al., 2008; Masiello et al., 2012; Liuzzi et al., 2014); a similar approach has been used by Rowe et al. (2006) and Rowe and Walden (2009). An estimate of the profile uncertainty based on a comparison with radiosonde profiles is given in Tab. 1 and derived in Sect. 6.3.

### 4.3 Columns of $O_3$, $CO_2$, $CH_4$, and $N_2O$

Total columns of ozone are obtained from nearby Brewer-Dobson soundings at the nearby Hohenpeissenberg observatory of the German Weather Service (Köhler, 1995) with an accuracy of ~1 % (Steaehelin et al., 2003). The horizontal distance between Hohenpeissenberg (47.80 °N, 11.02 °E, 985.5 m a.s.l.) and the Zugspitze is ~40 km and the altitude difference was taken into account by correcting the measured ozone columns by a factor of 0.982 inferred from the ozone profile of the (midlatitude winter) US standard atmosphere.

Column-averaged mixing ratios of carbon dioxide, methane, and nitrous oxide ($XCO_2$, $XCH_4$, $XN_2O$) were inferred from solar FTIR measurements. One option is to use the Zugspitze solar FTIR which is at the same time used for the NIR radiance measurements (see Fig. 1). However, for practical reasons (beamsplitter change from KBr to $CaF_2$ necessary for switch between MIR and NIR trace gas measurements, but not possible via remote control) the NIR FTIR instrument operated at the nearby Garmisch site (47.48 °N, 11.06 °E, 743 m a.s.l.) within the Total Carbon Column Observing

Network (TCCON; www.tccon.caltech.edu) has been used for routine operations. The horizontal distance between Garmisch and Zugspitze is only ~8 km. The site altitude difference has been taken into account for $CH_4$ and $N_2O$ because of the strospheric slope of the mixing ratio profiles of these species. This has been performed by using the multi-annual mean ratio of column averaged mixing ratios retrieved from the Zugspitze and Garmisch NDACC solar FTIR measurements of 1.8 % (the underlying datasets are displayed in Fig. 1 of Sussmann et al., 2012). Uncertainties given in

Tab. 1 were taken from the TCCON wiki (https://tccon-wiki.caltech.edu/Network_Policy/Data_Use_Policy/Data_Description#Sources_of_Uncertainty).

### 4.4 Aerosol optical depth

Aerosol optical depth (AOD) can become comparably high to water vapor continuum optical depth in NIR window regions and thereby potentially hinder accurate continuum quantification from field experiments as pointed out e.g. by Ptashnik et

al. (2015). Therefore AOD in the NIR has to be constrained precisely. For this we use sun photometer measurements in 12 channels between 339–1640 nm of the SSARA instrument developed by the Meteorological Institute of the University of



Munich (Toledano et al., 2009) and set up at Schneefernerhaus (2675 m a.s.l., 680 m horizontal distance to the Zugspitze solar FTIR). Our AOD retrieval is based on Toledano et al. (2009) and outlined in detail in Part III. We utilize radiance measurements in 5 channels from 440 to 780 nm and extrapolate AOD to the NIR via a fit of the Ångström relation. The corresponding uncertainties given in Tab. 1 are derived in detail in Part III.

## 5 Radiative transfer calculations

Synthetic radiance spectra in the Zugspitze closure experiment were generated using the LBLRTM radiative transfer model (Clough et al., 2005). The atmospheric state necessary as an input to the model was set according to the measurements listed in Sect. 4. Parameters not constrained by measurements were set to the values given by the midlatitude winter US standard atmosphere. For spectral line parameters, the aer_v3.2 line list provided alongside the LBLRTM model was used.

The calculations were carried out for a 39-level atmosphere from observer height (2964 m a.s.l.) to 120 km altitude. The altitude grid was chosen in order to keep the error from discretization of the atmosphere in the calculations negligible compared to the remainder of the residual error budget (2.8 % of total uncertainty for water vapor continuum retrieval windows). Synthetic radiance spectra were convoluted with a sinc-type instrumental line shape accounting for the $OPD_{max}$ relevant for the E-AERI (see Knuteson et al., 2004b) and solar FTIR (see Sect. 3.2) measurements, respectively.

## 6 Uncertainty analysis of radiance residuals

A meaningful interpretation of the spectral residuals derived in the closure experiment relies on a comprehensive residual uncertainty budget. For this purpose, systematic and 2-$\sigma$ statistical error estimates were set up for all significant individual uncertainty contributions. Radiance uncertainties were then calculated from input parameter uncertainties by multiplying them with the corresponding radiance derivatives. In the case of input profiles, state error covariance matrices were used. The radiance derivatives were calculated with the LBLRTM using the finite difference method, except for the $T$ profile radiance derivative matrix, which is calculated using the LBLRTM built-in analytic Jacobian capability.

### 6.1 Uncertainty from spectral radiance measurements

A first group of contributions to the uncertainty is associated with the AERI spectral radiance measurements. An estimate of the AERI measurement noise (Fig. 4 a) is automatically generated by the E-AERI software within the radiometric calibration procedure according to the method established by Revercomb et al. (1988). This noise estimate was reduced by 50 % to account for the effect of the PCA filter applied to the spectra (see Sect. 4). Further radiance uncertainty of the E-AERI measurements ensues from radiometric calibration errors. The calibration uncertainty estimate was set according to Knuteson et al. (2004b), who demonstrate this contribution to be less than 0.67 % (2-$\sigma$ uncertainty) of the ambient blackbody radiance. According to the same authors, the repeatability (precision) is 0.13 % (2 $\sigma$). The resulting absolute E-





AERI radiance uncertainty is shown in Fig. 4a via the purple line, which - divided by the grey ambient blackbody Planck curve – reflects the cited 0.67-% relative calibration uncertainty.

Uncertainty contributions associated with the NIR radiance measurements are the solar FTIR measurement noise and the radiometric calibration uncertainty. The calibration uncertainty includes sources of uncertainty connected with the temporal

stability of the calibration which are due to variation of the instrument's field of view on the solar tracker mirrors and ice buildup on the detector causing additional absorption. We show in Fig. 5 the overall 2-$\sigma$ calibration uncertainty (purple) which is between 0.6-1.7 % of measured radiance. For a plot of individual contributions we refer to Part II (Fig. 6 therein).

## 6.2 Uncertainty from radiative transfer calculations

The second group of contributions to the residual uncertainty is associated with the synthetic spectra calculation and the

corresponding input for spectroscopic line parameters and atmospheric state. A further uncertainty contribution associated with the LBLRTM ensues from discretization of the atmosphere used for the calculation. As outlined in Sect. 5, the layering was adjusted in order to keep the discretization error negligible compared to the remainder of the uncertainty budget.

## 6.3 Uncertainty from atmospheric state measurements

The uncertainties in IWV in case of FIR and MIR closure experiments based on E-AERI spectra are derived in Appendix A. For the FIR closure data set (detailed in Sect. 7), a mean IWV precision of 4.3 % (2-$\sigma$) is achieved, while the mean IWV bias is 4.4%. The resulting IWV related radiance uncertainty is shown in Fig. 4 (blue).

In the case of the NIR closure using solar FTIR spectra, the uncertainty of the IWV retrieval (precision: 0.8 %, bias: 1.1 %) is provided in Schneider et al. (2012). The IWV-related radiance uncertainty in the NIR is shown in Fig. 5 (blue).

In addition to the total water vapor column, erroneous input for the shape of the water vapor profiles from NCEP leads to errors in the synthetic radiance. A conservative estimate for this was inferred from a comparison of the NCEP profiles with radiosonde measurements. We used radiosonde data from a campaign performed close to the Zugspitze site between Mar – Nov 2002 (for details see Sussmann and Camy-Peyret, 2002, 2003; Sussmann et al., 2009). The campaign data set comprises a number of 181 pairs of radiosondes launched with a 1-hour time separation, and each radiosonde pair has been

combined to a best estimate of the state of the atmosphere according to the formalism by Tobin et al. (2006). Subsequently, both NCEP profiles and sonde-based Tobin-best-estimate profiles were normalized by IWV analogous to the analysis in the later closure experiment (Sect. 7), and then profile differences were computed. The red line in Fig. B1 shows the mean difference profile. The profile shape bias of 1.7 % given in Tab. 1 is just a simple proxy calculated from the mean of the moduli of the difference profile vector components. The statistical profile shape uncertainty was setup via an error

covariance matrix constructed from the difference profiles between NCEP and sonde-based Tobin-best-estimate profiles.



This error covariance was used for the further statistical analysis of radiance uncertainty. Just to illustrate some properties of this covariance, the black error bars in Figure B1 show the 2-$\sigma$ statistical uncertainties of the difference profile (corresponding to the diagonal of the covariance). By calculating the mean of these error bars we can derive a simple scalar proxy for the statistical profile uncertainty of 9.4 % (Tab. 1). Radiance uncertainties were then computed from the profile

uncertainty contributions by multiplying these with the corresponding derivative matrix of radiance with respect to water vapor profile shape (see Fig. B2). This leads to the residual uncertainty shown in Fig. 4 (pink).

The temperature profiles used in the closure study are a composite of $T$ profiles retrieved from the E-AERI spectra for the altitude range between the Zugspitze up to ~3.5 km a.s.l. and a NCEP resimulation profile at higher altitude as described in Sect. 4. The uncertainty estimate for these composite profiles was constructed from the same radiosonde campaign data as

for the water vapor profile analysis outlined above. To generate an estimate of the uncertainty, synthetic radiance spectra were calculated using all radiosonde-derived best-estimate $T$ profiles from the campaign. The systematic part of the uncertainty was estimated by adding the E-AERI calibration bias (0.66 %, see Tab. 1) to the synthetic radiance spectra. Then, the near-surface temperature profile retrieval described in Sect. 4 was applied to the modified radiances. Finally, the differences between our composite $T$ profiles and the radiosonde-based best-estimate profiles from the campaign were

calculated (red line in Fig. B3). Note, that the sign of the bias below 3.5 km a.s.l. (see Fig. B3) is arbitrary in the sense that it depends on whether the calibration bias is added or subtracted. The random uncertainty of the composite $T$ profile was estimated by adding random error according to the statistical E-AERI calibration uncertainty (0.13 %, Tab. 1) and E-AERI noise (yellow line in Fig. 4) to the synthetic radiance spectra. Finally, the near-surface temperature profile retrieval described in Sect. 4 was applied to the modified radiances. An error covariance matrix estimate was then calculated from

the difference of the radiosonde profiles to these composite $T$ profiles. Radiance uncertainties were then calculated by multiplication with the corresponding radiance derivative matrix depicted in Fig. B4. The resulting overall radiance uncertainties are shown in Figs. 4 and 5 (green).

Line parameter uncertainties for water vapor and further trace gases were set according to the error codes given in the aer_v3.2 line list provided alongside the LBLRTM radiative transfer model. The uncertainty of each parameter was

assumed to correspond to the mean of the error range specified by the error code value. Column uncertainties of further trace gases (see Tab. 2) are given by the TCCON specifications in the case of $CO_2$, $CH_4$ and $N_2O$ and the combined Brewer-Dobson measurement uncertainty for $O_3$. The resulting radiance uncertainties are depicted in Figs. 4 and 5 (red and cyan).

An additional contribution ensues in the NIR from the AOD uncertainty, which is <0.0015 at 2500 cm$^{-1}$ and <0.0025 at

7800 cm$^{-1}$ at airmass 1 as detailed in Part III. The resulting radiance uncertainty is shown Fig. 5 (grey).



## 6.4 Total uncertainty budget

Figure 4 shows an estimate of the residual uncertainty in the FIR and MIR closure experiment using AERI spectra; the same is shown in Fig. 5 for the solar FTIR radiative closure experiment in the NIR. The individual uncertainty contributions presented in Sect. 6.1–6.3 were added in quadrature to obtain the total residual uncertainty.

Figure 4a shows that the dominant contribution to the total uncertainty in the FIR is from water vapor line parameters, while $T$-profile uncertainties dominate in the MIR. Exceptions from this overall tendency do exist and are shown in Fig. 4b as an example for the FIR where a dominant role of $T$-profile uncertainties can be seen within saturated regions, e.g. around 420 cm$^{-1}$. However, such saturated regions are not included in the spectral micro-windows used for continuum quantification (Fig. 4c).

Uncertainty contributions to NIR radiance shown in Fig. 5 are dominated by varying contributions depending on wavelength. The overall uncertainty is dominated by water vapor line parameter uncertainties in the vicinity of the strong water vapor bands. The window regions in between are dominated by solar FTIR calibration uncertainties at low NIR wavenumbers but uncertainties due to AOD become an increasing and dominant contribution towards higher NIR wavenumbers. Exceptions are methane or nitrous oxide bands in the NIR, where the associate line parameter uncertainties

dominate the overall uncertainty.

## 7 Example closure study: assessment of FIR continuum

An example for a current research question that can be addressed with the closure setup presented in this publication is the magnitude of the water vapor continuum in the FIR spectral range. The Zugspitze closure experiment enables continuum quantification in the region $400 - 580$ cm$^{-1}$ based on a comparison of AERI radiance spectra and LBLRTM results (see Fig.

6a).

### 7.1 Spectra selection

The example analysis is based on measurements carried out in the Dec 2013 – Feb 2014 period. Several selection criteria were applied to the E-AERI measurements in order avoid bias in the quantification of the water vapor continuum. Clear-sky spectra were selected based on a radiance threshold in the MIR atmospheric window where significant thermal

emission occurs only under cloudy conditions. Namely, the mean radiance in the 829 to 835 cm$^{-1}$ window was required to be less than the synthetic radiance in this window plus the E-AERI calibration uncertainty presented in Sect. 6.1.

Due to the reduced number of suitable windows for continuum retrieval under moist atmospheric conditions, we selected only spectra with IWV < 5 mm. Snow accumulation on the LHATPRO may bias the measurements and can be detected based on the LHATPRO LWP measurements. Therefore, we only selected spectra with LWP < 100 g/m$^2$. NCEP reanalysis





data is used to constrain water vapor profile shape in the closure experiment. Despite the low uncertainties of the NCEP water vapor profiles demonstrated in Sect. 6.3, significant deviations from the real profile shape are expected in rare cases. This is due to the limited (6 hourly) time resolution of the NCEP data and its inability to reproduce small-scale spatial variability of water vapor concentrations. In order to identify these cases, we excluded measurements from further analysis

if the mean difference of NCEP vs. LHATPRO water vapor profiles exceeded the 1-$\sigma$ uncertainty of the LHATPRO measurements presented in Sect. 6.3. These criteria lead to a continuum retrieval data set of 211 spectra, selected from 2787 spectra measured in Dec 2013 – Feb 2014.

### 7.2 Window selection

Spectral residuals, i.e. the difference between synthetic and measured spectra were calculated from the set of selected
spectra. Figure 6b shows the mean residuals for our data set and their uncertainty according to the estimate provided in Sect. 6.

Accurate constraints on the water vapor continuum can only be derived from a number of spectral windows, whereas throughout the remainder of the spectrum the continuum does either not contribute significantly to the measured radiance or the residual uncertainty is too high. In order to select suitable windows, an estimate of the continuum uncertainty
achievable in the closure experiment was determined by multiplying the residual uncertainty estimate (see Sect. 6) with the continuum Jacobian. The continuum Jacobian, i.e. the derivative of continuum magnitude with respect to measured downwelling radiance, was calculated via the finite difference method using the MT_CKD 2.5.2 model as a priori. We selected windows for further analysis for which the continuum uncertainty is less than 100% above the minimum uncertainty in 10 cm$^{-1}$-wide bins. The selected windows are highlighted in red in Fig. 6b.

### 7.3 Continuum quantification procedure

Continuum quantification is achieved via an iterative minimization of spectral residuals in the selected windows. Spectral residuals in the windows are interpreted to be due to inaccurate foreign continuum since the radiance contribution by the self continuum is assumed to be negligible given the spectral range and the dry atmospheric conditions. Mean adjusted continuum coefficients are calculated in 10 cm$^{-1}$-wide bins to reduce influence of measurement noise and ILS uncertainty
on the results.

The individual analysis steps comprise a determination of the spectral residuals in the selected windows and subsequent adjustment of the continuum according to these results and the continuum Jacobian. Synthetic radiance is then recalculated using the adjusted continuum input. This process is repeated iteratively until the mean spectral residuals in all bins are below 10% of the residual uncertainty estimate.





## 7.4 Results

Figure 7 shows the mean foreign continuum coefficients determined from the Zugspitze data set in 10 cm$^{-1}$-wide bins. Appendix C contains our results in tabulated form. The results are fully consistent with the MT_CKD 2.5.2 model given the continuum uncertainty estimate according to Sect. 6. As visible in Fig. 7, our results are also fully consistent with the findings of Liuzzi et al. (2014) that were obtained in a comparable atmospheric closure study carried out in Antarctica.

## 8 Summary and conclusions

After a review of the state of the art in quantifying water vapor radiative processes we have detailed the instrumental setup of the new Zugspitze long-term radiative closure field experiment designed to cover the terrestrial and solar infrared between 400 to 7800 cm$^{-1}$ (1.28-25 µm). As a benefit for such experiments, the Zugspitze mountain site frequently encounters atmospheric states with very low IWV (minimum = 0.1 mm, median = 2.3 mm) and very low AOD (0.0024-0.0032 at 7800 cm$^{-1}$ at airmass 1). We also provided an uncertainty estimate for all measurements and retrieval procedures.

Given the instrumental uncertainties we assessed the sensitivity of the field experiment with respect to the information attainable, e.g. on the water vapor continuum. This was performed by setting up a systematic residual radiance uncertainty budget for the radiative closure over whole spectral range of the experiment. The dominant uncertainty contribution in the FIR is from water vapor line parameters, while $T$-profile uncertainties dominate in the MIR. Exceptions from this overall tendency do exist, e.g. for the FIR where a dominant role of $T$-profile uncertainties is found within saturated regions. However, such saturated regions are not included in the spectral micro-windows used for continuum quantification. Uncertainty contributions to NIR radiance residuals are dominated by varying contributions depending on wavelength. The overall uncertainty is dominated by water vapor line parameter uncertainties in the vicinity of the strong water vapor bands. The window regions in between are dominated by solar FTIR calibration uncertainties at low NIR wavenumbers, but uncertainties due to AOD become an increasing and dominant contribution towards higher NIR wavenumbers. Exceptions are methane or nitrous oxide bands in the NIR, where the associated line parameter uncertainties dominate the overall uncertainty.

Finally, we showed a water vapor continuum quantification in the FIR spectral region (400–580 cm$^{-1}$) and detailed all procedures involved, like spectral micro-window and data quality selection. The FTIR foreign continuum coefficients determined from the Zugspitze data set are consistent both with the MT_CKD 2.5.2 model and the recent atmospheric closure study carried out in Antarctica by Liuzzi et al. (2014).

Two companion papers Part II (Reichert et al., this issue) and Part III (Reichert and Sussmann, this issue) will show details on the development of a radiometric calibration of the Zugspitze solar FITR system for NIR radiance measurements and its application to derive first information on the NIR water vapor continuum under atmospheric conditions.





Future work aims at extending our studies from water vapor radiative closure to also include a quantification of the radiative properties of cirrus clouds. Because of the regionally varying radiative properties of cirrus it is important to perform such studies at various field sites around the globe.

**Appendix A: Retrieval of IWV from E-AERI spectra**

**A.1 Retrieval method**

We utilize an approach similar to the method proposed by Serio et al. (2008), i.e. IWV is retrieved via a derivative approach using one iteration to minimize E-AERI vs. LBLRTM spectral residuals in IWV-sensitive windows. As first guess IWV, data from a LHATPRO microwave radiometer are used. LHATPRO (Radiometer Physics, Germany; Rose et

al., 2005), designed for ultra-low humidity sites (IWV < 4.0 mm), is a microwave radiometer located side-by side to the E-AERI.

The procedure for selection of suitable spectral windows for IWV retrieval from the $400 – 600$ $cm^{-1}$ spectral range has been implemented as follows:

i) All regions in which continuum has significant influence on the downwelling radiance (relative continuum uncertainty <

100%) are excluded in order to avoid biased water vapor continuum quantification results due to the IWV fit.

ii) The uncertainty of the IWV fit for single spectral points is calculated for the remaining windows. IWV relative uncertainty is given as the residual uncertainty excluding IWV contribution divided by $\partial\boldsymbol{I}/\partial$IWV, i.e. the derivative of downwelling spectral radiance $\boldsymbol{I}$ with respect to IWV. The overall uncertainty comprises two classes of errors, namely type-i errors which are uncorrelated with wavenumber, and type-ii errors correlated with wavenumber.

iii) Spectral points (channels) are ordered from lowest to highest type-ii uncertainty.

iv) Ensembles with stepwise increased number of channels are constructed including channels with increasing type-ii uncertainty, and the overall uncertainty (type i + ii) is calculated for each ensemble. Figure A1 shows this overall uncertainty depending on the number of included channels. E-AERI measurement noise and line parameter errors are treated as type-i error contributions (the underlying assumption being that line parameter errors for different lines are

independent). Therefore, these contributions to the cumulative uncertainty are reduced by a factor $1/\mathrm{sqrt}(n)$ when $n$ channels are included in the fit (causing the decrease of uncertainty on the left hand side of Fig. A1). All other uncertainty contributions (E-AERI calibration, T profile errors, and water vapor profile errors, see Sect. 6.3 for details) are correlated for different spectral channels, therefore no uncertainty reduction is achieved by including more channels in the fit, and the overall uncertainty increases toward the right hand side of Fig. A1. This is because more and more channels with increasing

type-ii uncertainty are included.





v) The optimum number of spectral channels for the fit is deduced from the minimum of overall (type-i + ii) uncertainty (Fig. A1). The resulting optimum numbers of channels for the different spectra of our closure dataset are shown in Fig. A2; the mean value is 23.7 channels, with a minimum of 8 and a maximum of 40 channels.

The results of the IWV fit for all spectra included in the FIR closure data set are shown in Fig. A3. The mean correction relative to the LHATPRO first guess IWV was -0.051 mm, with a standard deviation of 0.075 mm.

### A.2 Uncertainty estimate

An estimate of the statistical and systematic uncertainty of the IWV retrieval can be obtained based on the uncertainty of the E-AERI-LBLRTM spectral residuals presented in Sect. 6 and Fig. 4.

The statistical residual uncertainty is calculated as the quadratic sum of the AERI measurement noise and the statistical uncertainties related to calibration, $T$ profiles, and water vapor profiles. The IWV fit uncertainty for single spectral points ensues as the statistical residual uncertainty divided by $\partial I/\partial$IWV. However, the IWV fit result is not derived from single spectral points but from an ensemble of points selected according to the criterion presented in Fig. A1. Therefore, the statistical IWV uncertainty for each spectrum results as the error-weighted mean of the single-point-contributions for all channels included in the ensemble. The mean statistical IWV uncertainty we achieve for the FIR closure data set (Sect. 7) is 4.3 % (2-$\sigma$).

The systematic IWV uncertainty can be derived in an analogous way. Systematic error contributions due to line parameters, E-AERI calibration, $T$ profiles, water vapor profiles, and columns of further species are summed up quadratically to calculate the systematic residual uncertainty. Using the same further analysis as outlined above for the statistical contribution, a mean systematic IWV uncertainty of 4.4 % ensues for the FIR closure data set.

### Appendix B: Errors in the shape of water vapor and temperature profiles – supplementary figures

*Acknowledgments.* We thank H.P. Schmid (KIT/IMK-IFU) for his continuous interest in this work and institutional support to purchase the LHATPRO. This project was funded by the Bavarian State Ministry of the Environment and Consumer Protection via grants TLK01U-49581 and VAO-II TPI/01. Andreas Reichert received a PhD grant of the Deutsche Bundesstiftung Umwelt (DBU). The authors are indebted to D.D. Turner (NOAA) and E. Mlawer (AER) for helpful conversations during the definition phase of the project; in particular, it is a pleasure to thank E. Mlawer for his suggestion to utilize the Zugspitze solar FTIR instrument for near-infrared radiative closure studies. We are grateful to P. Hausmann (KIT/IMK-IFU who performed the solar FITR water vapor retrievals. Thanks to U. Köhler (German Weather Service) for providing Brewer-Dobson data and M. Wiegner (LMU) for giving access to sun photometer measurements.



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





## Tables

**Table 1.** Instruments and geophysical parameters measured at the Zugspitze radiative closure experiment. Uncertainties are given for 2-$\sigma$ confidence.

| geophys. Parameter | instrument | repeat cycle | uncertainty/specification |
|---|---|---|---|
| FIR & MIR spectral radiance ($400 - 3000$ cm$^{-1}$) | E-AERI | 10 min | [1]resolution 0.5 cm$^{-1}$ |
| | | | calibration bias <0.66 % of ambient BB radiance |
| | | | calibration precision <0.13 % of ambient BB radiance |
| NIR spectral radiance ($2500 - 7800$ cm$^{-1}$) | solar FTIR | $75 - 150$ s | [1]resolution 0.011 cm$^{-1}$ |
| | | | calibration accuracy 0.6-1.7 % of measured radiance |
| IWV (E-AERI) | retrieval from E-AERI spectra | 10 min | bias 4.4 % |
| | | | precision 4.3% |
| IWV (solar FTIR) | retrieval from solar FTIR spectra | $75 - 150$ s | bias 1.1 % |
| | | | precision 0.8 % |
| water vapor profile shape | NCEP | 6 h | bias 1.7 % |
| | | | precision 9.4 % |
| temperature profile | E-AERI & NCEP | 10 min | accuracy <1 K |
| O$_3$ column | Brewer-Dobson | ~30 min | accuracy <1 % |
| XCO$_2$ | TCCON | 100 s | bias <0.07 % |
| | | | precision <0.25 % |
| XCH$_4$ | TCCON | 100 s | bias <1.04 % |
| | | | precision <0.3 % |
| XN$_2$O | TCCON | 100 s | bias <1.85 % |
| | | | precision <0.5 % |
| NIR AOD | SSARA | 1 s | accuracy at airmass 1 |
| | | | <0.0015 (@ 2500 cm$^{-1}$) |
| | | | <0.0025 (@ 7800 cm$^{-1}$) |

[1]resolution defined as 1 / maximum optical path difference (OPD$_{max}$)



**Table 2.** Climatological statistics of clear-sky IWV levels above the Zugspitze derived from $N$ multi-annual solar FTIR measurements shown in Fig. 3. Numbers are given in units of (mm).

| $N$ | mean | stdv | min | median | max |
|---|---|---|---|---|---|
| 7388 | 3.0 | 2.2 | 0.1 | 2.3 | 12.0 |

5  **Table C1.** Mean foreign continuum coefficients derived from the Zugspitze closure measurements and associated (2-$\sigma$) uncertainties.

| wavenumber [cm$^{-1}$] | $c_f$ [cm$^2$/(cm$^{-1}$ molec)] |
|---|---|
| 407.06 | $1.91 \cdot 10^{-25} \pm 3.60 \cdot 10^{-26}$ |
| 411.49 | $1.79 \cdot 10^{-25} \pm 3.46 \cdot 10^{-26}$ |
| 435.46 | $1.27 \cdot 10^{-25} \pm 3.09 \cdot 10^{-26}$ |
| 447.40 | $9.09 \cdot 10^{-26} \pm 2.69 \cdot 10^{-26}$ |
| 464.67 | $8.00 \cdot 10^{-26} \pm 1.90 \cdot 10^{-26}$ |
| 478.01 | $5.90 \cdot 10^{-26} \pm 1.59 \cdot 10^{-26}$ |
| 488.02 | $5.24 \cdot 10^{-26} \pm 1.30 \cdot 10^{-26}$ |
| 495.83 | $4.47 \cdot 10^{-26} \pm 1.03 \cdot 10^{-26}$ |
| 512.96 | $3.71 \cdot 10^{-26} \pm 1.04 \cdot 10^{-26}$ |
| 525.27 | $2.88 \cdot 10^{-26} \pm 1.05 \cdot 10^{-26}$ |
| 534.45 | $2.73 \cdot 10^{-26} \pm 7.02 \cdot 10^{-27}$ |
| 543.45 | $2.26 \cdot 10^{-26} \pm 7.84 \cdot 10^{-27}$ |
| 555.80 | $2.00 \cdot 10^{-26} \pm 5.94 \cdot 10^{-27}$ |
| 562.63 | $1.76 \cdot 10^{-26} \pm 6.10 \cdot 10^{-27}$ |
| 573.54 | $1.70 \cdot 10^{-26} \pm 6.84 \cdot 10^{-27}$ |
| 585.15 | $1.10 \cdot 10^{-26} \pm 6.65 \cdot 10^{-27}$ |



**Figures**

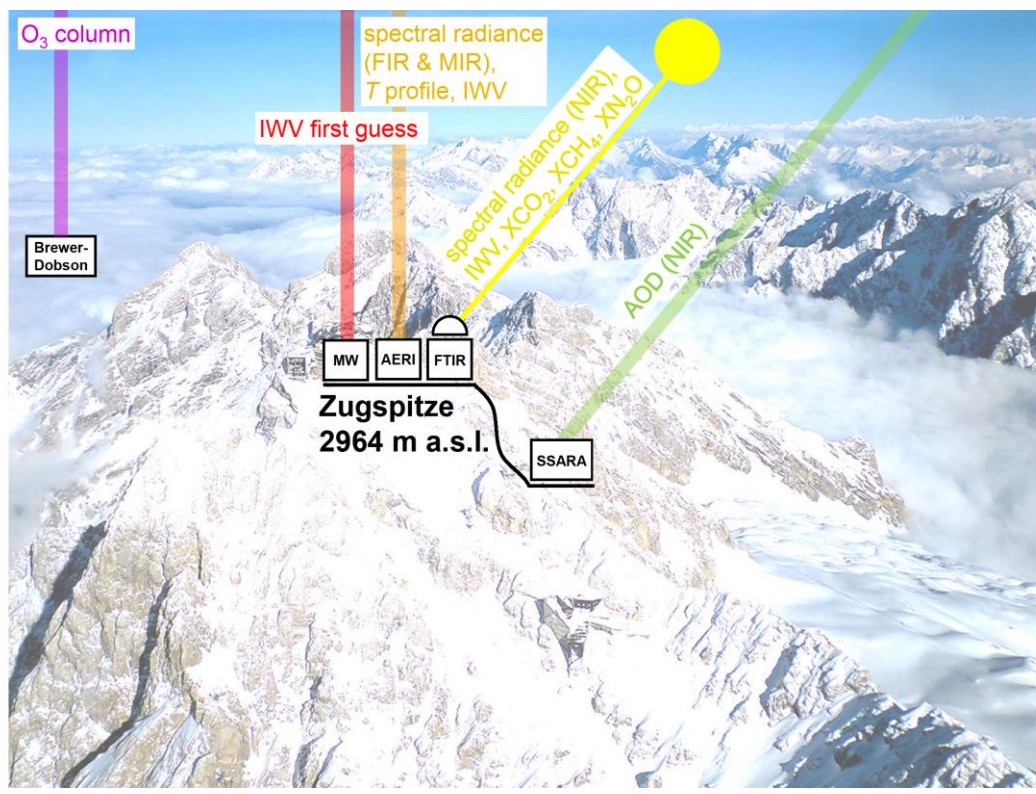

**Figure 1.** Instrumental setup of the Zugspitze radiative closure experiment covering the FIR, MIR, and NIR spectral range (FTIR: Bruker
IFS 125 HR high-resolution solar Fourier-Transform Infrared Spectrometer; AERI: extended-range Atmospheric Emitted Radiance
Interferometer; MW: LHATPRO low-humidity microwave radiometer; SSARA: sun photometer; Brewer-Dobson: ozone
spectrophotometer).





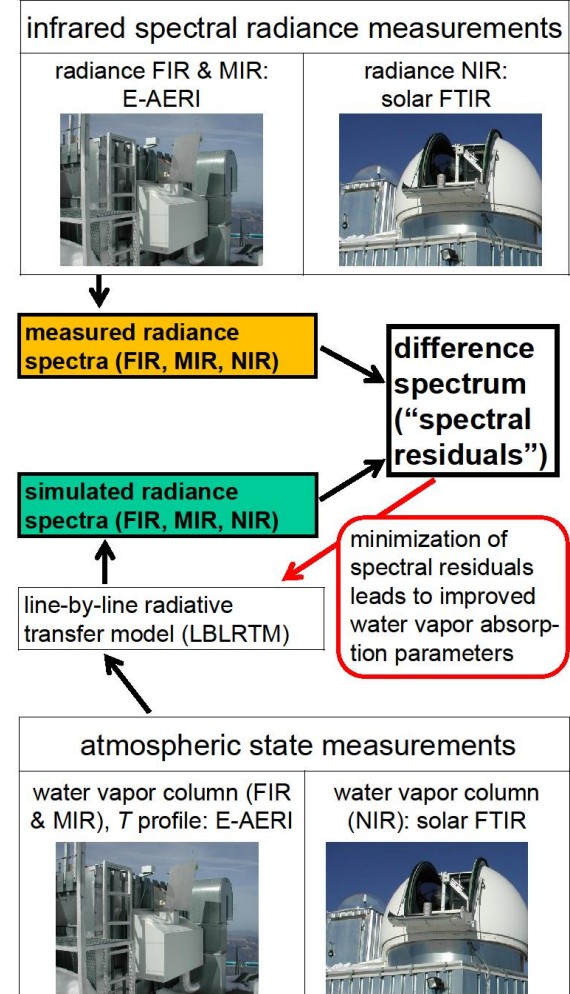

**Figure 2.** Logical scheme of the Zugspitze radiative closure experiment. Simulated radiance spectra are based on atmospheric state measurements performed coincidently to the radiance measurements. The closure idea is to minimize spectral residuals between simulated and measured radiance spectra by iteratively adjusting/improving the water vapor absorption parameters used in the FIR, MIR, and NIR spectral radiance simulations.





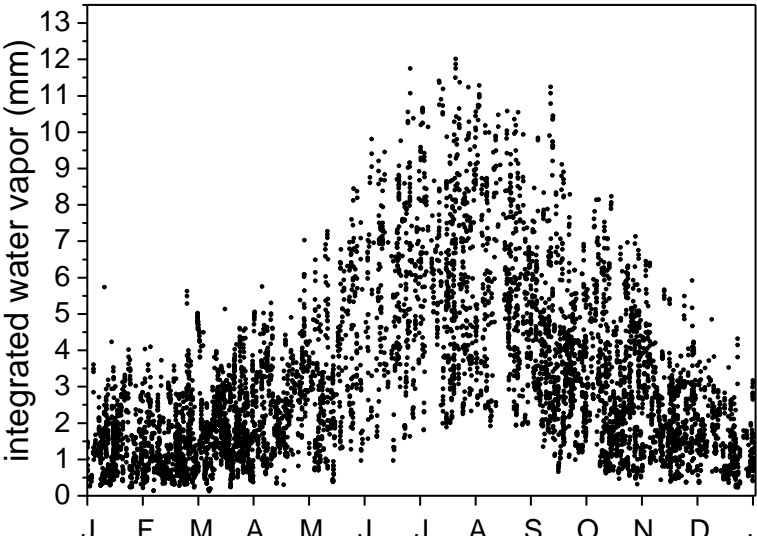

**Figure 3.** Climatology of integrated water vapor above the Zugspitze. Data are from multi-annual (1996 – 2013) Zugspitze solar FTIR measurements (clear sky, 15-20 min integration; see Sussmann et al., 2009 for details). See Table 1 for related statistics.





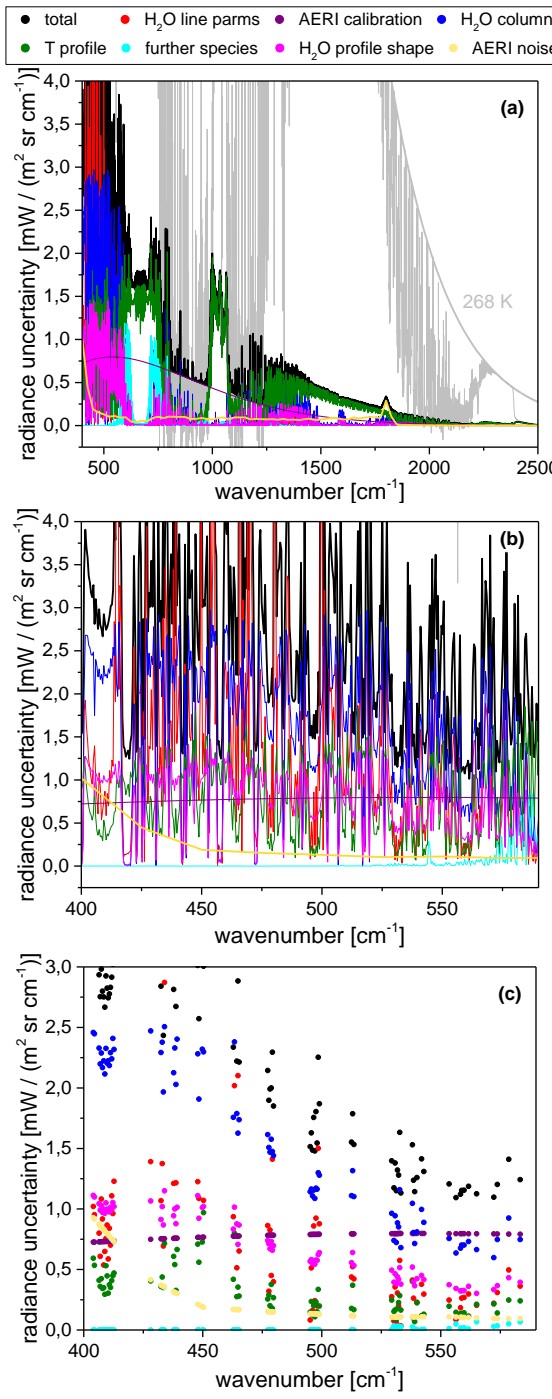

**Figure 4.** Residual uncertainty (2-$\sigma$, relative to ambient BB radiance) of the FIR and MIR closure experiment for a single AERI thermal emission spectrum and for the mean atmospheric state of the closure data set (IWV = 1.6 mm, for more details see Section 7.1). **(a)** Individual error contributions (colors) to the total residual uncertainty (black). For reference, a calculated radiance spectrum (grey) for





the mean atmospheric state is shown along with the ambient BB Planck curve. **(b)** Zoom of a) for the FIR part. **(c)** Same as b) but restricted to the spectral windows used for continuum quantification.

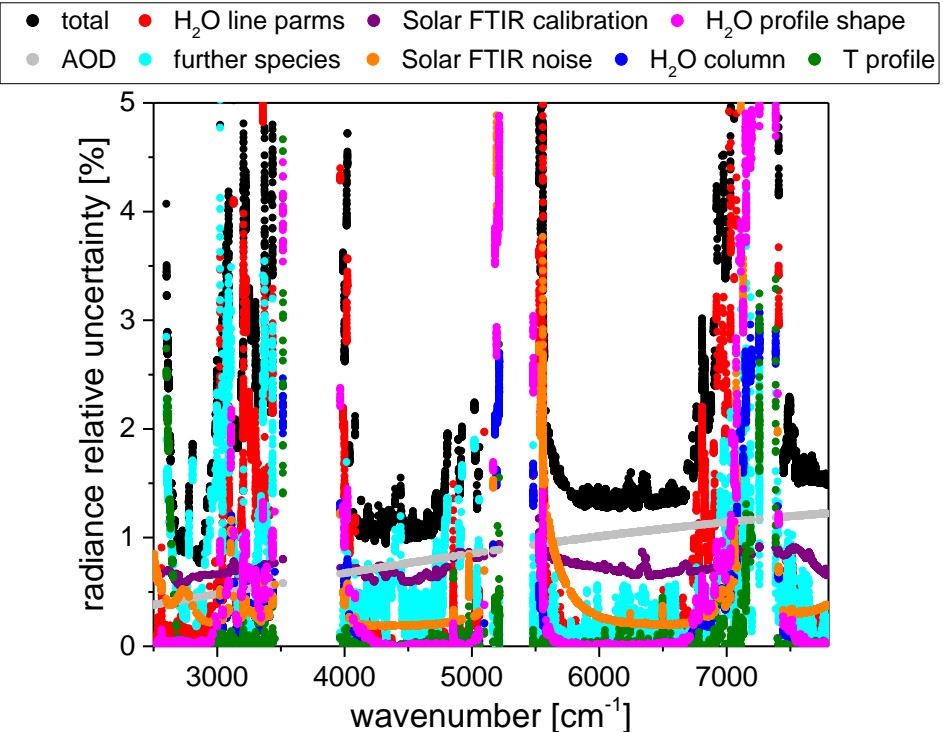

**Figure 5.** Residual uncertainty (2-σ, relative to measured solar radiance) of the NIR closure experiment for a single solar FTIR spectrum and for the mean atmospheric state of the closure data set (IWV = 2.3 mm, for more details see Part III). The total residual uncertainty (black) results from contributions by water vapor line parameter uncertainties (red), IWV uncertainty (blue), temperature profile uncertainty (green), further trace gas column and line parameter uncertainties (cyan), AOD uncertainty (grey), solar FTIR calibration

10 uncertainty (purple) and solar FTIR measurement noise (orange). Representation corresponds to the mean atmospheric state of the water vapor continuum quantification data set and the spectral windows used for continuum retrieval.





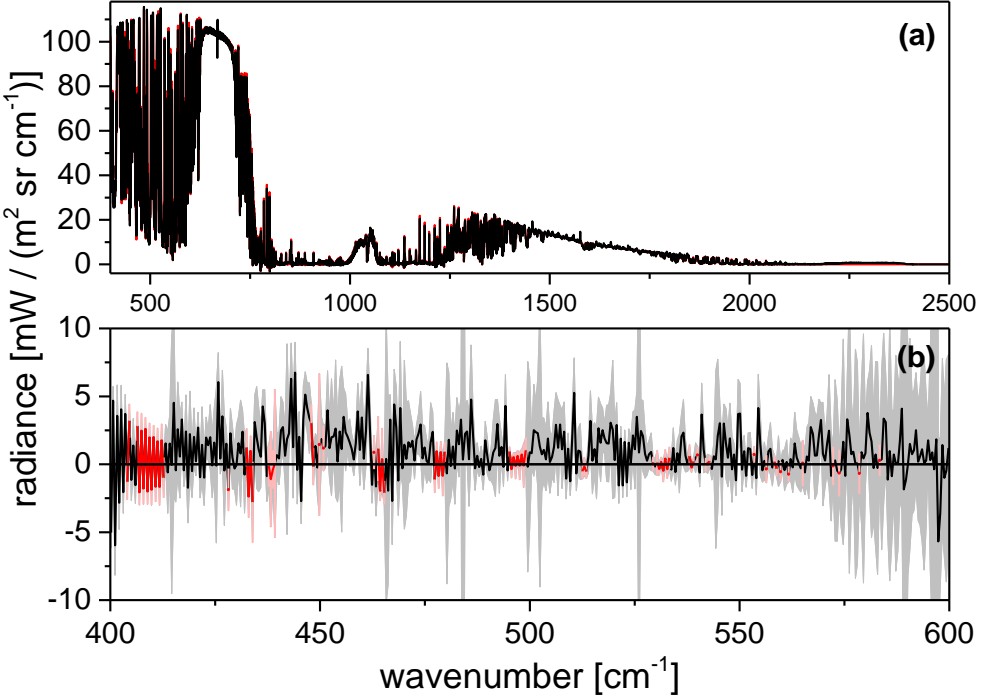

**Figure 6. (a)** Comparison of a measured AERI spectrum (black) recorded on 13 Dec 2013, 8:28 UTC and the corresponding synthetic LBLRTM spectrum (red). **(b)** Mean spectral residuals derived from the continuum retrieval data set (black) and residual uncertainty (grey). Spectral windows used for continuum retrieval are highlighted in red.

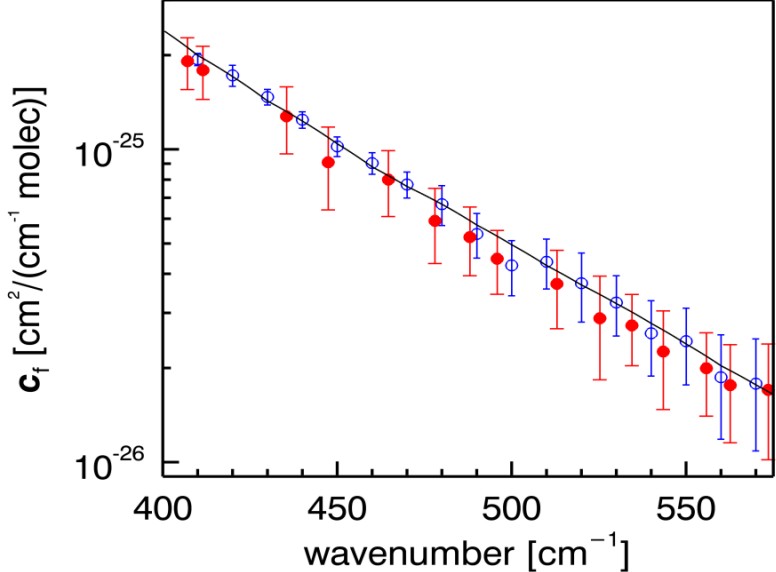

**Figure 7.** Mean foreign continuum coefficients derived from the Zugspitze closure measurements (red) in comparison to the MT_CKD 2.5.2 model (black) and the results of Liuzzi et al. (2014) (blue).





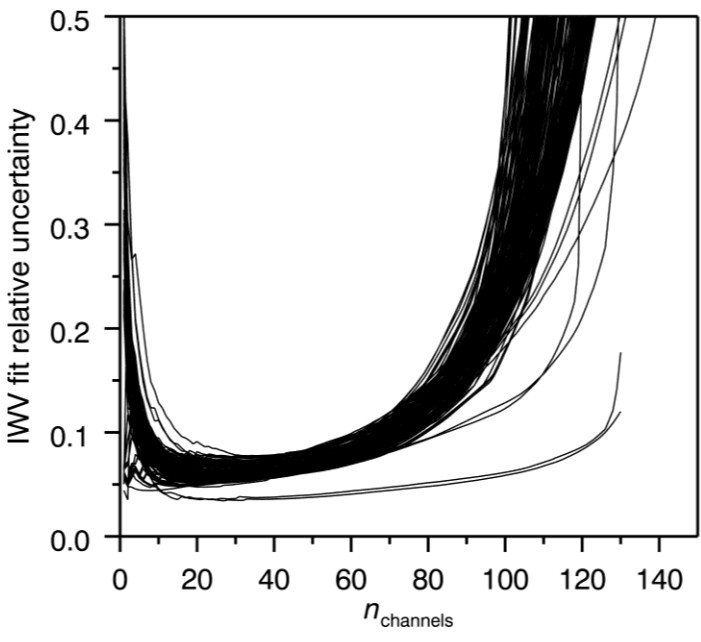

**Figure A1.** Relative uncertainty of the IWV fit depending on the number of spectral points $n_{channels}$ used in the fit for the spectra included in the FIR continuum data set. Channels are ordered by increasing type ii-uncertainty. The number of channels used for the fit is adjusted in order to yield minimum overall uncertainty.

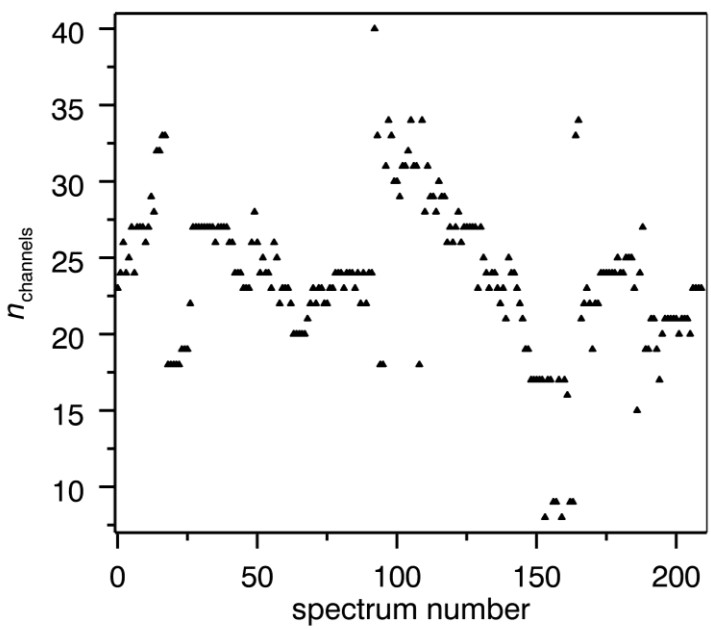

**Figure A2.** Number of spectral channels included in the IWV fit for the spectra of the FIR continuum data set. $n_{channels}$ was chosen according to the minimum uncertainty criterion shown in Fig. A1.





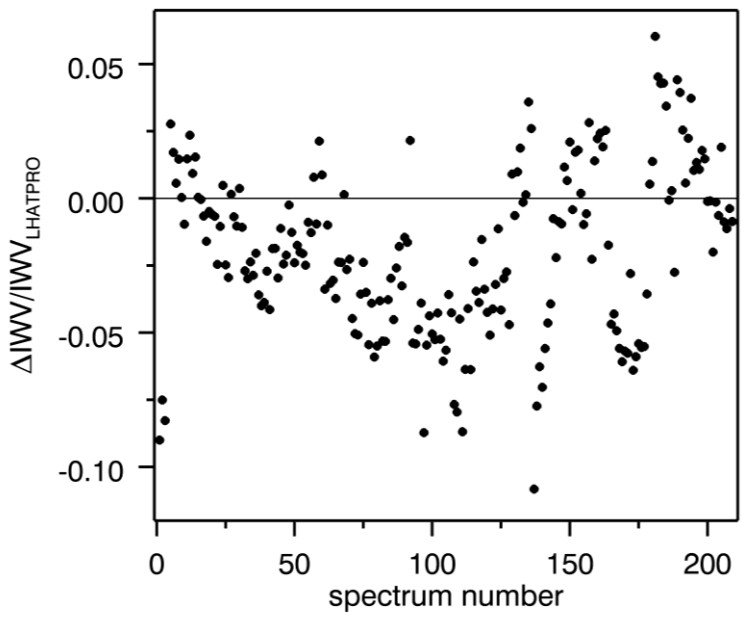

**Figure A3.** Relative adjustment $\Delta IWV/IWV_{LHATPRO}$ derived in the IWV fit for the spectra included in the FIR continuum data set.





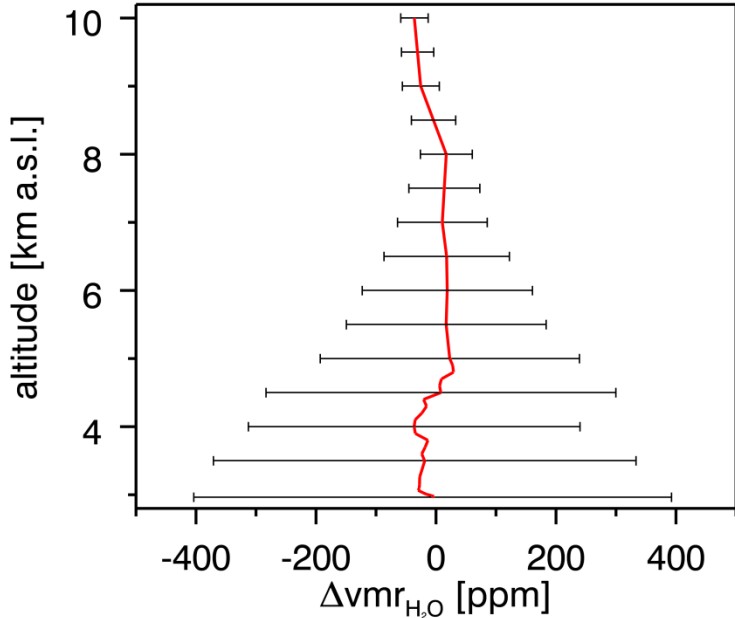

**Figure B1.** Uncertainty analysis of NCEP water vapor profile shape. Red is the mean difference between NCEP profiles normalized with respect to IWV and an ensemble of best-estimate profiles derived from pairs of radiosondes launched with a 1-h separation (also normalized for IWV). Black error bars indicate 2-$\sigma$ differences.

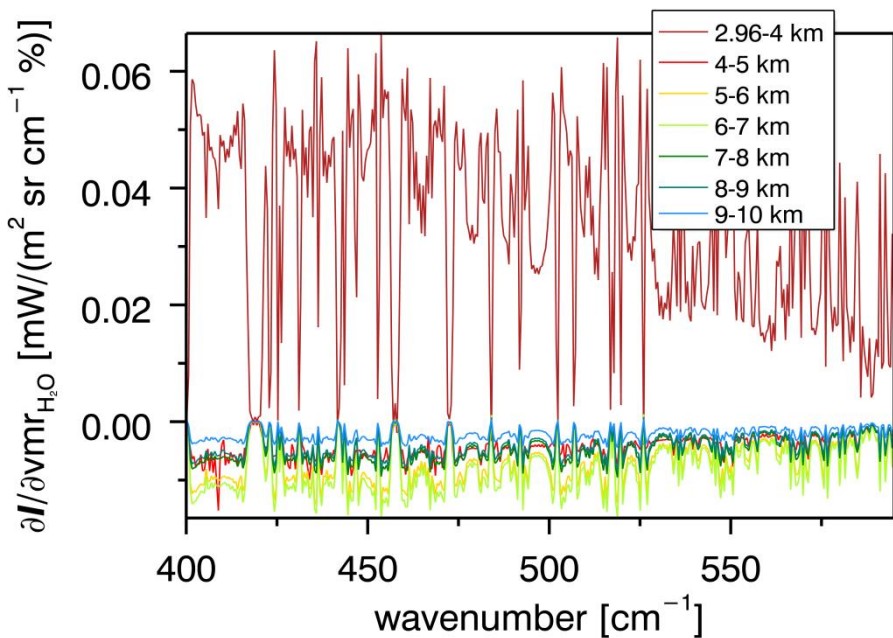

**Figure B2.** Derivative of surface downwelling radiance with respect to water vapor profile shape computed for the mean atmospheric state of the continuum retrieval data set. Color coding indicates the contributions from different altitude layers.





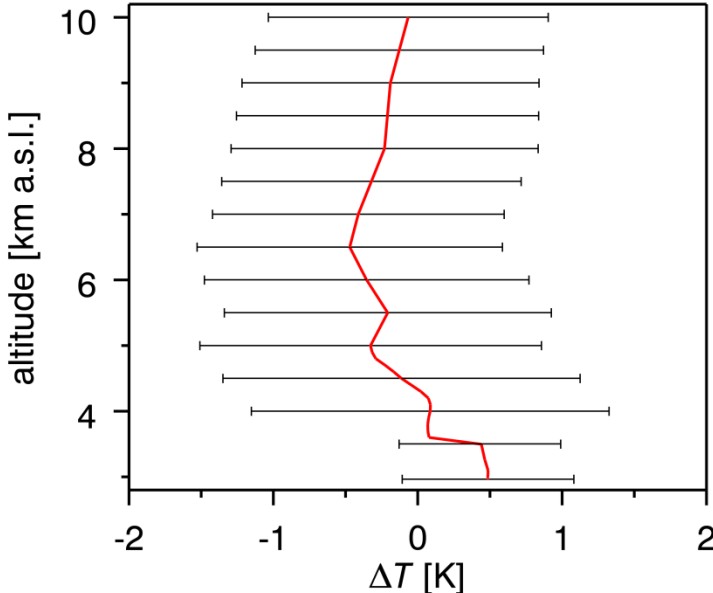

**Figure B3.** Uncertainty analysis of $T$ profiles used in the closure experiment (composite of E-AERI retrievals < 3km and NCEP). Red is the mean difference between these composite profiles and an ensemble of best-estimate profiles derived from pairs of radiosondes launched with a 1-h separation. Black error bars indicate 2-$\sigma$ of the differences.

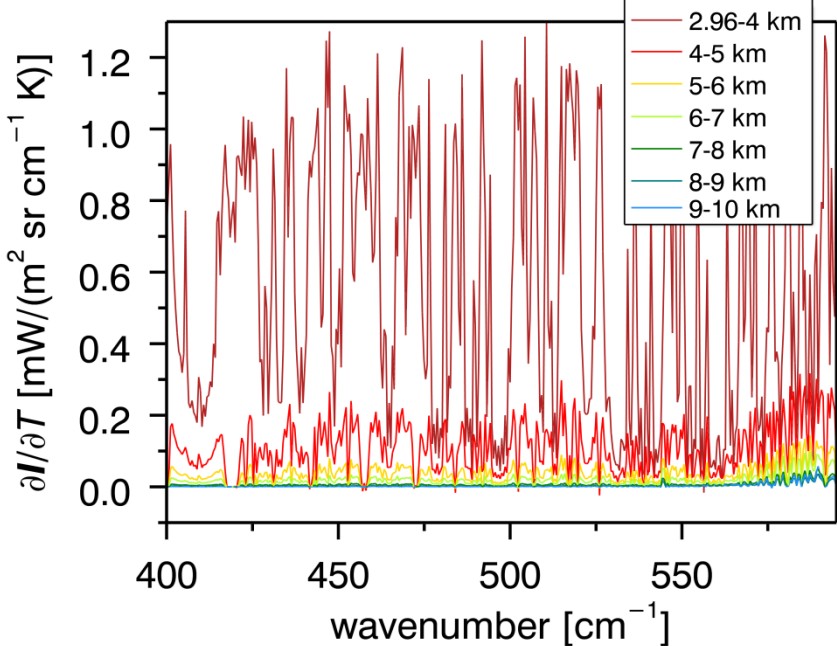

**Figure B4.** Derivative of surface downwelling radiance with respect to the $T$ profile computed for the mean atmospheric state of the continuum retrieval data set. Color coding indicates the contributions from different altitude layers.