# Peer review of "The Zugspitze radiative closure experiment for quantifying water vapor absorption over the terrestrial and solar infrared. Part I: Setup, uncertainty analysis, and assessment of far-infrared water vapor continuum"

_Atmospheric Chemistry and Physics, 2016_

## Referee Comment (RC1) · Anonymous Referee #1 · 27 May 2016

The paper describes the setup of an E-AERI instrument on top of mount Zugspitze. The desired operation time of the instrument is about 10 years and it will be used to perform a water vapour enclosure experiment. In contrast to other studies this experiment is not campaign based, but will perform continuous long-term measurements of the FIR and MIR spectral range. The high altitude of the stations enables to measure under low AOD conditions and low IWV concentration (down to about 0.2 mm). Such measurements are needed for the improvement of the water vapor spectroscopy used

in remote sensing and climate predictions. An interesting new aspect of this study is that NIR measurements are used for the water vapour enclosure experiment. Those are performed by a Bruker FTIR spectrometer on mount Zugspitze.

The paper is well written and gives an extensive outline of the current literature. The initial setup of the instrument is described together with an error analysis and a practical example of the concept (estimation of the water vapour continuum in the FIR). The error analysis of the atmospheric state is well performed, which is in my opinion a crucial point of the study. Since knowledge of water vapour spectroscopy is limited, measurements like this are still needed. The paper is in the scientific scope of ACP and I recommend publication of the work after minor revision.

Some comments and questions:

page 31, section 5: I understand that in the retrieval spectral windows are selected with a minimal continuum contribution to estimate the IWV. Then in a second step the water vapour continuum is estimated from spectral regions with strong and weak continuum contributions.

a) When you would correct your spectra for the estimated water continuum as described in your manuscript, does that mean that it then would be possible to retrieve IWV concentrations from retrieval windows with different water vapor continuum contributions in agreement with the IWV estimated from the ones with the weak contribution only? How strong would the IWV difference be in that case?

b) Is it not possible to retrieve the IWV column together with the water vapour continuum per iteration step? I think using such an approach would mean that you could use wider retrieval windows and would gain a better error estimation of the IWV and the water vapour continuum, since the interference between those two would then be included in the error estimation.

page 10, line 26: You are performing a PCA filter to reduce the noise on the spectra.

Can you tell something about the statistics of the residuum for multiple spectra? Is it fairly normal distributed and what would then be the value of sigma?

page 29, Fig. 4 a) and b): Here too many lines are plotted on top of each other. Please find a better representation. In the current state it is very difficult to distinguish the different contributions. Further the grey line in Fig. 4a) is not mentioned in the legend.

page 30, Fig. 5: Like in Fig. 4: Here to many lines are plotted on top of each other. Maybe it would be better to make subplots?

---

## Referee Comment (RC2) · Anonymous Referee #2 · 13 Jun 2016

This paper details an experimental setup of a long-term radiative closure experiment on the Zugspitze mountain site. The special environmental conditions (frequent low IWV & AOD) coupled with the setup of the Zugspitze instruments provide an advantageous way of characterizing the sensitivity of the radiative closure over the whole infrared range. This study and the following companion papers are major contributions for the improvement of water vapor spectroscopy used in atmospheric remote sensing techniques and climate simulations. The paper is well written and the authors provide

a comprehensive analysis of the past and current developments in this field.

Important points and questions have already been raised by Referee #1. I agree and recommend the publication of this work on ACP after minor revisions.

Technical corrections:

Page 5, line 19: add comma after "these are crucial prerequisites for closure studies and on dry winter days"

page 5, line 19: change " the Zugspitze offers regularly" to "the Zugpsitze regularly offers"

page 5, line 23: change "Network of the Detection…" to "Network for the Detection…"

page 5, line 24: add comma after "(NDACC; www.ndacc.org)"

page 9, lines 5-7: consider restructuring the sentences to: "We use the retrieval scheme developed by Esposito et al. (2007) for this kind of boundary layer temperature inversion, which has been successfully utilized by a series of studies (Serio et al., 2008; Masiello et al., 2012; Liuzzi et al., 2014). A similar approach has been used by Rowe et al. (2006) and Rowe and Walden (2009).

Figures 4&5, As referee #1 already mentioned, too many lines are plotted on top of each other such that the contributions are difficult to distinguish. May I also suggest to not use red and green colors together.

---

## Referee Comment (RC3) · Anonymous Referee #3 · 28 Jun 2016

This paper presents a nice description of the set-up at Zugsptize and an application of the observations at this site to determine FIR H2O continuum coefficients. The paper will be a useful reference to the observations at this station and make a solid addition to the literature on the experimental determination of the H2O continuum. We will recommend publication after improvements are made to this manuscript, especially those marked "major" below (which constitute the "specific comments" in this review).

pg 2

[Figure]

17 – Some species have collision-induced absorption so this isn't strictly true.

20 – "RRTM" is the full name of the code

23 – "aliased" is not the proper word. Maybe "which potentially introduces biases into applications ..."

25 – "e.g." should not begin a sentence

30 – Why not say H2O-air and leave out the "mainly"?

30 – Sounds better with "still a definite continuum theory does not exist..." 3 1 – It is probably not true that a consensus has been reached that both processes contribute appreciably, so it would be better to say "two possible physical processes".

pg 3

2- The wording here should be thought through more carefully. The foreign continuum would be a "dimer" of an H2O molecule with an air molecule, so perhaps not strictly a "water dimer".

3 – The Pfeilsticker et al. result is not viewed as very credible. Perhaps reference here the Ptashnik paper and another CAVIAR paper.

7-8 – I think MT_CKD is slightly different than as described here. The model is built off of a sum of monomer lines, but the "collision-induced term" (it's more appropriate to refer to it as in the Mlawer et al 2012 paper as due to a "weak interaction") is a presumed collisional complex between a monomer and another molecule, perhaps more of a quasi-stable complex.

13 – MT_CKD coefficients are uncertain everywhere, so perhaps say "are more uncertain".

14 – don't begin sentence with "e.g."

16 – 19 – This is a pretty old study. Also consider discussing the results of d'Angelis et

al. 2015

pg 4

5 – "measurement', not "measure"

18 – Since the RHUBC-I results are so relevant to this study, it makes sense to list that campaign too 1 9-20 – Mlawer et al. presented such a closure experiment at the 2014 HITRAN meeting.

(major) 19 through pg5, 9 – This is one of many places in the paper where details pertinent to the NIR analysis are provided. These places detract from the focus of this paper, which is on the set-up at Zugspitze, which pertains to all experiments, and the FIR spectroscopic studies. These many text sections should not be in this paper, but in the one about the NIR analysis. Restrict mention of non-FIR material to aspects of the instrumental set-up at Zugspitze.

pg 5

10 – "maturate" is not a word. Perhaps "advance".

pg 7

4 and elsewhere – this instrument is abbreviated "ER-AERI" by its developers

7 – the regular AERI (not the ER) was used in RHUBC-II in the Atacama

10 – front end is two words

11 – "... and two blackbodies.." This part of the sentence is poorly written.

22 – Remove "for numbers"

pg 9

10-15 – What O3 profile was scaled to to agree with the column measurement? MLW? Was it truncated below the Zugspitze altitude to get the 0.982 factor? This is unclear.

14 – The MLW and US standard are different profiles.

15-26 – This explanation should be improved. The phrase "has been used for routine operations" is particularly unclear.

22 – Is "stropheric" the correct term?

28 – Change "comparably high" to "comparable".

Sec 4.4 – The authors should move this to the paper on the NIR. This material is not really relevant in this paper.

pg 11

3-7 – Again, this NIR information should not be in this paper.

9-14 – The material covered on pg 12, lines 23-28, should be in this section.

20 through pg 12, line 6 –

There are a number of aspects that could be improved about this section: 1) it is odd that Appendix B has only figures and no text. If these figures were part of a supplemental section, then this might be an acceptable space saver, but it doesn't seem like that is the case. Either move the figures to the main text or move the text with the details about this analysis to Appendix B. 2) there is no context to understand Figure B1 since the reader doesn't know if 200 ppm is a large or small percentage of the total $H_2O$ abundance. Could a second panel be added to that plot with the average $H_2O$ profile? 3) Fig. B2 is hard to interpret. Is what's plotted the change in radiance for a one percent change in that layer's $H_2O$? I'm guessing the sign is different for the first layer (at least I think it's the first layer – the two red colors are hard to distinguish) due to temperature inversions? It seems strange that the magnitudes of the derivatives for the first and second layers are so different since the $H_2O$ amounts probably aren't that different. 4) From the text, it seems that the $H_2O$ profile uncertainty shown in Figure 4 is not simply due to the diagonal term in the covariance matrix but the layer-to-layer

correlations are taken into account. Is this correct? If so, it's unclear from the text how the math works. 5) ln 27 – Change "later closure experiment (Sect. 7)" to "the closure experiment described in Section 7" (assuming that's what is meant). 6) ln 29 – "set up" is 2 words 7) 28-29 – "mean of the moduli of the difference profile vector components" is hard to understand

pg 12

8 – What is "resimulation profile"? Assimilation?

7-22 – Was there no met tower at the site with a direct temperature measurement? Was the AERI T retrieval up to 3.5 km limited to just opaque $CO_2$ spectral regions? Was the uncertainty in the retrieval itself (a posteriori) accounted for in this analysis? What about the uncertainty in the AERI temperature retrieval due to spectroscopic uncertainty? (major)

23-28 – Using the line parameter uncertainty codes is likely to cause a significant underestimation of the actual uncertainty. These codes come from HITRAN and aren't necessarily reliable. It is recommended that the authors get better estimates of the the uncertainty by comparing values in recent databases, such as HITRAN 2008 vs. HITRAN 2012 (and for widths, the values in Delamere et al.). For widths, differences of $\sim$20% are common – it might be reasonable to assume that for all lines in this uncertainty analysis. Also, given the low temperatures at this site, the uncertainty in the temperature dependence of the widths should also be accounted for, and it is unclear if it has. These values have had some very large changes from HITRAN 2008 to HITRAN 2012.

pg 13

5-9 – It is puzzling that the supposed dominant role of the $H_2O$ line parameters is mentioned first when, for the regions of interest in terms of continuum derivation, the dominant uncertainty is the $H_2O$ column (as shown in blue in Fig 4c). This paragraph

should be reworded to emphasize the key conclusions of the uncertainty analysis as it pertains to the continuum.

29 – It's unclear what the threshold of LWP < 100 has to do with snow accumulation on the LHATPRO.

pg 14

10 – Figure 4c indicates that the total uncertainty in the continuum channels between 400-500 cm-1 range from 2-3 radiance units. However, in Figure 6b it doesn't look like the underlying gray uncertainty is that far from the red parts of the residual curve, especially from 470-500 cm-1. Also, please define the residual to be obs-calc or calc-obs.

pg 16 (major) The authors have chosen to obtain the H2O column by a retrieval in the FIR, which presents the possibility of circularity since the same instrument, uncertain line parameters, etc. are being used the the column determination and the continuum derivation. Also, the continuum itself is an element of the column determination. Clearly the authors must believe that this provides a better estimation of the column than alternative sources for the column. Similar closure studies (e.g. Turner et al., Delamere et al.) have derived the column from microwave measurements near lines that have line parameters with low uncertainty, removing potential circularity and lowering a key source of uncertainty. No details are given for the column retrieval by the LHATPRO, but it may use a similar approach as used in these other closure studies. It would be interesting for the authors to provide the rationale for their choice for determining the column. Why do they feel it provides a better value than the microwave? How different are the column values obtained from each approach? Is this difference a good estimate of the uncertainty in the column amount they are using? A plot should be provided with these differences. What is the method used in the LHATPRO retrieval?

The definition and description of type-i and type-ii uncertainty (bottom of page) should be moved up to points ii or iii, instead of its current placement in point iv.

(major) The uncertainty due to the line parameters is likely underestimated here. First, as above, the width errors should not be obtained from the error code in the line file. Second, the temperature dependence of the widths need to be accounted for (assuming they are not). Lastly, comparing HITRAN 2012 to the aer line file that is used in this study shows that the width differences are not equally likely to be positive as negative – the signs of the differences are usually the same. Therefore, assuming that the resulting uncertainties are uncorrelated between different spectral regions is not appropriate. Although this type of correlation between different spectral regions may appear to be more accidental than other correlated errors (e.g. T profile), since the widths tend to come from calculations that are constrained to observed values, there may be a clear reason why they would generally be high (or low) in a particular version of the database. Reclassifying the line parameter errors as correlated (or somewhere in between correlated and uncorrelated) would increase the uncertainty in the column estimation.

19 – It is better to refer to this as "uncorrelated between wavenumbers" and "correlated between wavenumbers" rather than the current wording.

pg17

11, 19 – "ensues" is not the correct word

pg 25 Table C1 contains key results and should be moved from an appendix into the main part of the paper

---

## Author Response (AR1)

**Author Response, Andreas Reichert, Karlsruhe Institute of Technology, Garmisch-Partenkirchen, Germany, 20 August 2015**

Dear Dr. Maring,

a point-by-point response to the reviews to our manuscript acp-2016-321 is given below and available online at http://www.atmos-chem-phys-discuss.net/acp-2016-321/acp-2016-321-AC1-supplement.pdf. We thank the anonymous referees for their sound and very constructive revision that greatly helped to improve our manuscript. A marked-up version of the manuscript highlighting all changes made by the authors is also attached to this response.

We are confident that all referee comments have been addressed thoroughly and hope that a final publication in ACP is possible soon.

Sincerely,

Andreas Reichert, 30 August 2016

**Author response to the referee comments:**

We thank the anonymous referees for their very sound, constructive and helpful comments which helped us to significantly improve our manuscript. In the following, we provide point-to-point replies to all comments made by the referees. All page and line numbers quoted in this reply refer to the initial version of the manuscript.

**Anonymous Referee #1**

**Comments and questions:**

**Page 31, section 5:** I understand that in the retrieval spectral windows are selected with a minimal continuum contribution to estimate the IWV. Then in a second step the water vapour continuum is estimated from spectral regions with strong and weak continuum contributions.

a) When you would correct your spectra for the estimated water continuum as described in your manuscript, does that mean that it then would be possible to retrieve IWV concentrations from retrieval windows with different water vapor continuum contributions in agreement with the IWV estimated from the ones with the weak contribution only? How strong would the IWV difference be in that case?

*Repeating the IWV retrieval with the newly derived continuum leads to a mean IWV difference of about 2%. While this difference is well below the IWV uncertainty given in the initial manuscript, it indicates a small interference between the IWV fit and the continuum retrieval. To avoid this interference, we decided to adopt the strategy proposed by the referee in point b).*

*Using this strategy of repeating the IWV fit iteratively, the mean difference of IWV results between the window selection outlined in the initial manuscript and an alternative selection including also points with strong continuum contribution is negligible (0.005 mm). This means that using the iterative fit method, the IWV fit can be extended to windows with significant continuum contributions, which has been implemented in the updated manuscript (see also reply to point b).*

b) Is it not possible to retrieve the IWV column together with the water vapour continuum per iteration step? I think using such an approach would mean that you could use wider retrieval windows and would gain a better error estimation of the IWV and the water vapour continuum, since the interference between those two would then be included in the error estimation.

*We thank the referee for suggesting this very helpful alternative method to avoid interference between IWV fit and continuum retrieval. The proposed method was implemented in our data analysis and included in the description of the IWV fit method. The manuscript was adapted as follows (Page 17, line 3):*
*"v) The IWV fit according to steps i) - iv) is repeated for each iteration step of the continuum quantification procedure (see Sect. 7.3). This iterative approach serves to avoid interference between the continuum quantification and the IWV fit. Performing the IWV fit including only windows with negligible continuum contribution (i.e. excluding all windows with continuum uncertainty < 100 %) leads to a mean bias in the IWV results of 0.005 mm. This negligible bias indicates that the iterative approach is able to avoid significant interference between IWV fit and water vapor continuum determination."*

**Page 10, line 26:** You are performing a PCA filter to reduce the noise on the spectra. Can you tell something about the statistics of the residuum for multiple spectra? Is it fairly normal distributed and what would then be the value of sigma?

*The following text was added to the manuscript (Page 10, line 26):*
*"The residuals, i.e. the radiance component identified as noise by the PCA filter is well represented by a normal distribution (mean = $8.5 \cdot 10^{-6}$ mW/(m$^2$ sr cm$^{-1}$), σ = 0.21 mW/(m$^2$ sr cm$^{-1}$) for the closure data set presented in Sect. 7.1)"*

**Page 29, Fig. 4 a) and b):** Here too many lines are plotted on top of each other. Please find a better representation. In the current state it is very difficult to distinguish the different contributions. Further the grey line in Fig. 4a) is not mentioned in the legend.

*We thank the referee for pointing out the missing legend item, which was added to the revised figure. Figure 4 was subdivided in additional subplots to avoid too much data to be shown in each plot.*

**Page 30, Fig. 5:** Like in Fig. 4: Here to many lines are plotted on top of each other. Maybe it would be better to make subplots?

*Two subplots were included in the revised manuscript to improve the representation of Fig. 5.*

**Anonymous Referee #2**

**Technical corrections:**

**Page 5, line 19:** add comma after "these are crucial prerequisites for closure studies and on dry winter days"

*The manuscript has been changed as suggested by the referee.*

**Page 5, line 19:** change " the Zugspitze offers regularly" to "the Zugspitze regularly offers"

*The manuscript has been changed as suggested by the referee.*

**Page 5, line 23:** change "Network of the Detection…" to "Network for the Detection…"

*The manuscript has been changed as suggested by the referee.*

**Page 5, line 24:** add comma after "(NDACC; www.ndacc.org)"

*The manuscript has been changed as suggested by the referee.*

**Page 9, lines 5-7:** consider restructuring the sentences to: "We use the retrieval scheme developed by Esposito et al. (2007) for this kind of boundary layer temperature inversion, which has been successfully utilized by a series of studies (Serio et al., 2008; Masiello et al., 2012; Liuzzi et al., 2014). A similar approach has been used by Rowe et al. (2006) and Rowe and Walden (2009).

*The manuscript has been changed as suggested by the referee.*

**Figures 4&5**: As referee #1 already mentioned, too many lines are plotted on top of each other such that the contributions are difficult to distinguish. May I also suggest to not use red and green colors together.

*Figures 4 and 5 were divided in additional subfigures enable the reader to fully distinguish different contributions. The simultaneous use of red and green colors was avoided.*

**Anonymous Referee #3**

**Page 2**
**17** – Some species have collision-induced absorption so this isn't strictly true.

*The manuscript was changed to avoid the misleading statement in the initial text:*
*"Modeling the radiative impact of the gas phase molecular compounds has to include radiative processes such as pure rotational absorption/emission in the far infrared (FIR) and vibration-rotation absorption/emission in the mid-infrared (MIR) and the near infrared (NIR)."*

**20** – "RRTM" is the full name of the code

*The erroneous term "Rapid Radiative Transfer Model" was removed from the manuscript.*

**23** – "aliased" is not the proper word. Maybe "which potentially introduces biases into applications ..."

*The manuscript was changed as suggested by the referee.*

**25** – "e.g." should not begin a sentence

*Page 2, line 25 was changed to: "For example, ..."*

**30** – Why not say H2O-air and leave out the "mainly"?

*The manuscript was changed as suggested by the referee.*

**30** – Sounds better with "still a definite continuum theory does not exist..."

*The manuscript was changed as suggested by the referee.*

**31** – It is probably not true that a consensus has been reached that both processes contribute appreciably, so it would be better to say "two possible physical processes".

*The manuscript was changed as suggested by the referee.*

**Page 3**
2- The wording here should be thought through more carefully. The foreign continuum would be a "dimer" of an H2O molecule with an air molecule, so perhaps not strictly a "water dimer".

*The wording was changed to (Page 3, line 2): " ii) dimer contributions, i.e. absorption due to stable and/or metastable dimers."*

**3** – The Pfeilsticker et al. result is not viewed as very credible. Perhaps reference here the Ptashnik paper and another CAVIAR paper.

*An additional reference to the CAVIAR-study by Ptashnik et al. (2011) was added to the manuscript as suggested.*

**7-8** – I think MT_CKD is slightly different than as described here. The model is built off of a sum of monomer lines, but the "collision-induced term" (it's more appropriate to refer to it as in the Mlawer et al 2012 paper as due to a "weak interaction") is a presumed collisional complex between a monomer and another molecule, perhaps more of a quasi-stable complex.

*The wording was changed to "weak interaction" as suggested.*

**13** – MT_CKD coefficients are uncertain everywhere, so perhaps say "are more uncertain".

*The wording was changed as suggested by the referee.*

**14** – don't begin sentence with "e.g."

*Page 3, line 14 was changed to "For example, "*

**16 – 19** – This is a pretty old study. Also consider discussing the results of d'Angelis et al. 2015

*A reference to De Angelis et al. (2015) was added to the discussion (Page 3, line 30):*
*"This result is consistent with the finding of DeAngelis et al. (2015) that the treatment of shortwave absorption by water vapor in climate models has a major influence on the response of the hydrological cycle to climate change."*

**Page 4**
**5** – "measurement', not "measure"

*The manuscript was corrected as suggested by the referee.*

**18** – Since the RHUBC-I results are so relevant to this study, it makes sense to list that campaign too

*RHUBC I was included in the list of campaigns in page 4, line 18 as suggested by the referee.*

**19-20** – Mlawer et al. presented such a closure experiment at the 2014 HITRAN meeting

*A reference to the study by Mlawer et al (2014) was added to the manuscript.*

(major) **19 through pg5, 9** – This is one of many places in the paper where details pertinent to the NIR analysis are provided. These places detract from the focus of this paper, which is on the set-up at Zugspitze, which pertains to all experiments, and the FIR spectroscopic studies.  These many text sections should not be in this paper, but in the one about the NIR analysis.  Restrict mention of non-FIR material to aspects of the instrumental set-up at Zugspitze.

**Page 11**
**3-7** – Again, this NIR information should not be in this paper.

**Sec 4.4** – The authors should move this to the paper on the NIR. This material is not really relevant in this paper.

*We thank the referee for pointing out that the NIR analysis was covered too extensively in the initial manuscript. The corresponding text was shortened significantly wherever possible in the revised manuscript to avoid overlap with the companion paper Part III and to not detract from the focus of this paper.*
*However, we respectfully disagree that the NIR analysis should be completely removed from the manuscript. The goal of our study, which is also expressed in the title, is to provide a description of the setup and a sensitivity analysis for the entire closure study, i.e. including the NIR part. In order to achieve this goal, we would like to keep a minimum of information on previous NIR work in the literature, our NIR instrumentation setup as well as the NIR sensitivity analysis is required in the manuscript.*

*Sect. 4.4 was shortened as follows: "Aerosol optical depth (AOD) is constrained using sun photometer measurements of the SSARA-Z instrument set up at Schneefernerhaus (2675 m a.s.l., 680 m horizontal distance to the Zugspitze solar FTIR). Our AOD retrieval and the derivation of the corresponding uncertainties given in Tab. 1 are outlined in detail in Part III."*

*Page 4, line 19 through page 5, line 9 was shortened as follows: "Coming to the NIR we note that for this spectral region to our knowledge no atmospheric radiative closure experiments have been reported in the literature with the exception of the studies by Sierk et al. (2004) and Mlawer et al. (2014). A hindrance for quantitative field studies may have been the fact that absorption in the NIR due to aerosols can become comparable to the magnitude of the water vapor continuum absorption of interest (Ptashnik et al., 2015). The possibility to accurately separate these two components depends on aerosol load (i.e. aerosol optical depth, AOD) and therefore on field site characteristics, as will be outlined when introducing the new Zugspitze field experiment below. On the other side, there have been many laboratory studies in the NIR range. Laboratory experiments using FTIR spectrometry and large cells have shown that the self- and foreign continuum within the windows was found to be significantly stronger than given by MT_CKD (Baranov and Lafferty, 2011; Ptashnik et al., 2011, 2012, 2013). Another issue is that laboratory measurements performed by different techniques have yielded to inconsistent results. For example, the magnitude of the self continuum in NIR windows derived from laboratory FTIR spectrometry is higher by about one order or magnitude compared to results obtained by cavity ring-down spectroscopy (CRDS; Mondelain et al., 2013, 2015), which furthermore significantly differ to laboratory results obtained by calorimetric interferometry (Bicknell et al., 2006). Finally, a drawback of laboratory measurements is that they are typically performed at least at room temperature or even heated, in order to detect the weak continuum absorption in the limited optical path length of the cells. Therefore, for climate and remote sensing applications an extrapolation of continuum coefficients to the lower atmospheric temperatures is required which may lead to significant inaccuracies due to the uncertainty of the self continuum temperature dependence (e.g. Shine et al., 2012)."*

**Page 5**
**10** – "maturate" is not a word. Perhaps "advance".

*The wording was changed as suggested by the referee.*

**Page 7**
**4** and elsewhere – this instrument is abbreviated "ER-AERI" by its developers

*The instrument abbreviation was changed to ER-AERI throughout the manuscript.*

**7** – the regular AERI (not the ER) was used in RHUBC-II in the Atacama

*The wording of the manuscript was changed (Page 7, line 6) to "AERI or ER-AERI instruments have been used..." to avoid the misleading statement that was given in the initial manuscript.*

**10** – front end is two words

*The manuscript was corrected as suggested by the referee.*

**11** – "... and two blackbodies.." This part of the sentence is poorly written.

*The wording was changed to (Page 8, line 11): "It comprises the scene mirror and two calibration blackbodies (BB), which are operated at ambient temperature and at 310 K, respectively (Fig. 2)."*

**22** – Remove "for numbers"

*The manuscript was changed as suggested by the referee.*

**Page 9**
**10-15** – What O3 profile was scaled to agree with the column measurement? MLW?
Was it truncated below the Zugspitze altitude to get the 0.982 factor? This is unclear.

*The wording was changed as follows to improve the description of the analysis (Page 9, line 12): "We used the ozone profile given by the midlatitude winter standard atmosphere, which was scaled to the measured total column corrected by a factor of 0.982. This correction is used to account for the altitude difference to the Zugspitze site and was deduced by calculating the fraction of the total ozone column between 985.5 m a.s.l. and 2964 m a.s.l. according to the MLW standard atmosphere. "*

**14 –** The MLW and US standard are different profiles.

*The correct term "midlatitude winter standard atmosphere" was used in the revised manuscript.*

**15-26** – This explanation should be improved.  The phrase "has been used for routine operations" is particularly unclear.

*The wording has been changed to provide a clearer outline of the measurements (Page 9, line 15):*
*" Column-averaged mixing ratios of carbon dioxide, methane, and nitrous oxide ($XCO_2$, $XCH_4$, $XN_2O$) were inferred from solar FTIR measurements. Trace gas column measurements can be obtained with the Zugspitze solar FTIR which is also used for the NIR radiance measurements in the closure experiment (see Fig. 1). However, for practical reasons (beamsplitter change from KBr to CaF2 necessary for switch between MIR and NIR trace gas measurements, but not possible via remote control), the NIR FTIR instrument operated at the nearby Garmisch site (47.48 °N, 11.06 °E, 743 m a.s.l.) within the Total Carbon Column Observing Network (TCCON; www.tccon.caltech.edu) has been used for routine  trace gas measurements. This is a suitable option, because the horizontal distance between Garmisch and Zugspitze is only ~8 km. The site altitude difference has been taken into account for CH4 and N2O because of the stratospheric slope of the mixing ratio profiles of these species. This has been performed by using the multi-annual mean ratio of column averaged mixing ratios retrieved from the Zugspitze and Garmisch NDACC solar FTIR measurements of 1.8 % (the underlying datasets are displayed in Fig. 1 of Sussmann et al., 2012). Uncertainties given in Tab. 1 were taken from the TCCON wiki (https://tccon-wiki.caltech.edu/Network_Policy/Data_Use_Policy/Data_Description#Sources_of_Uncertainty)."*

**22** – Is "stropheric" the correct term?

*The wording was corrected to "stratospheric".*

**28** – Change "comparably high" to "comparable".

*The manuscript was changed as suggested by the referee.*

**Page 11**
**9-14** – The material covered on pg 12, lines 23-28, should be in this section.

*The corresponding material was moved as suggested by the referee.*

**20 through pg 12, line 6** – There are a number of aspects that could be improved about this section:
1)  it is odd that Appendix B has only figures and no text. If these figures were part of a supplemental section, then this might be an acceptable space saver, but it doesn't seem like that is the case.  Either move the figures to the main text or move the text with the details about this analysis to Appendix B.

*Figures B1 – B4 were moved to the main text as suggested by the referee and re-labeled as Fig. 4 – 7 in the revised manuscript.*

2) there is no context to understand Figure B1 since the reader doesn't know if 200 ppm is a large or small percentage of the total H2O abundance. Could a second panel be added to that plot with the average H2O profile?

*A second panel showing the mean H2O profile was added to Fig. B1 (re-labeled as Fig. 4 in the revised manuscript) as suggested by the referee.*

3) Fig. B2 is hard to interpret. Is what's plotted the change in radiance for a one percent change in that layer's H2O? I'm guessing the sign is different for the first layer (at least I think it's the first layer – the two red colors are hard to distinguish) due to temperature inversions? It seems strange that the magnitudes of the derivatives for the first and second layers are so different since the H2O amounts probably aren't that different.

*The following explanation was added to the manuscript to facilitate the interpretation of Fig. B2, which was re-labeled as Fig. 5 in the revised manuscript (Page 12, line 6):*
*"The representation in Fig. 5 corresponds to the radiance change associated with a 1% change of water vapor density in a given altitude layer and subsequent rescaling of the profile to the IWV obtained as outlined in Sect. A.1. Due to the rescaling to a prescribed IWV, the 1%-increase of water vapor density in a given layer is associated with a decrease in all other layers. Therefore, a 1%-perturbation in the lowermost layer (2.96-4 km a.s.l.) corresponds to lowering the center of gravity of the water vapor profile and leads to a positive change in radiance, while for higher layers, the opposite is true. Due to the decrease of water vapor density with altitude (see Fig. 4b), the radiance effect of a 1%-perturbation decreases rapidly with altitude."*

4) From the text, it seems that the H2O profile uncertainty shown in Figure 4 is not simply due to the diagonal term in the covariance matrix but the layer-to-layer correlations are taken into account. Is this correct? If so, it's unclear from the text how the math works.

*The following outline of the radiance uncertainty calculation from the error covariance matrix is given in the revised manuscript (Page 12, line 4):*
*"An estimate of the corresponding radiance uncertainty that includes the influence of layer-to-layer correlations can be obtained by multiplying the full error covariance matrix with the*

*derivative matrix of radiance with respect to water vapor profile shape in the atmospheric layers (see Fig. 5) and its inverse."*

5) **ln 27** – Change "later closure experiment (Sect. 7)" to "the closure experiment described in Section 7" (assuming that's what is meant).

*The wording of the manuscript was changed as suggested by the referee.*

6) **ln 29** – "set up" is 2 words

*The manuscript was corrected as suggested by the referee.*

7) **28-29** – "mean of the moduli of the difference profile vector components" is hard to understand

*A more extensive explanation was provided to improve clarity (Page 11, line 28):*
*"The profile shape bias of 1.7 % given in Tab. 1 is just a simple proxy that has been obtained as follows: for each pair of sonde and NCEP profiles, a difference vector was calculated. Each component of the average bias vector was then deduced as the mean of the absolute values of the corresponding components of the difference vectors."*

**Page 12**
**8** – What is "resimulation profile"? Assimilation?

*A more precise wording was introduced at page 12, line 8:" …while at higher altitude the T profiles were set to according to the NCEP reanalysis"*

**7-22** – Was there no met tower at the site with a direct temperature measurement?
Was the AERI T retrieval up to 3.5 km limited to just opaque CO2 spectral regions?
Was the uncertainty in the retrieval itself (a posteriori) accounted for in this analysis?
What about the uncertainty in the AERI temperature retrieval due to spectroscopic uncertainty? (major)

*Unfortunately, no meteorological tower is available close to the Zugspitze AERI. The T retrieval was limited to the central part of the $CO_2$ band, namely $625 – 715$ $cm^{-1}$. Details on the wavenumber range selection are given in Esposito et al. (2007), which served as a template for our T retrieval scheme as outlined in the manuscript.*
*The uncertainty estimation scheme described in the manuscript relies on modifying synthetic radiance spectra according to the radiance uncertainty of our experiment and subsequently performing the T profile fit. Finally, the retrieved T profiles are compared to the input profile and to obtain the uncertainty estimate. This estimate therefore takes into account uncertainty contributions due to both a) the performance of the retrieval itself, e.g. smoothing effects caused by the retrieval, and b) possible radiance errors. To clarify this, the following text was added to the manuscript (Page 12, line 13): "This approach implicates that both the uncertainty due to the retrieval itself as well as additional uncertainty due to inaccurate radiance input are taken into account for the T profile uncertainty estimate."*
*The uncertainty due to spectral line parameters was included in the T profile error estimate. We thank the referee for pointing out that we failed to mention this important contribution in the initial manuscript. The manuscript was changed as follows (page 12, line 12): "The systematic part of the uncertainty was estimated by adding the ER-AERI calibration bias (0.66 %, see Tab. 1) and the estimated bias due to line parameter uncertainties (see Sect. 6.2) to the synthetic radiance spectra."*

**23-28** – Using the line parameter uncertainty codes is likely to cause a significant underestimation of the actual uncertainty. These codes come from HITRAN and aren't necessarily reliable. It is recommended that the authors get better estimates of the uncertainty by comparing values in recent databases, such as HITRAN 2008 vs. HITRAN 2012 (and for widths, the values in Delamere et al.). For widths, differences of 20 % are common – it might be reasonable to assume that for all lines in this uncertainty analysis. Also, given the low temperatures at this site, the uncertainty in the temperature dependence of the widths should also be accounted for, and it is unclear if it has. These values have had some very large changes from HITRAN 2008 to HITRAN 2012.

*As suggested by the referee, we have introduced the difference between HITRAN 2008 and HITRAN 2012 (updated with the results of Delamere et al., 2010) as an additional estimate for the line parameter uncertainties. However, this leads to underestimation of the uncertainties for lines where the parameters were not changed between HITRAN 2008 and HITRAN 2012. To obtain a conservative uncertainty estimate, we therefore use the maximum uncertainty value suggested by either the uncertainty codes or the HITRAN 2008 vs. HITRAN 2012 difference. The uncertainty in the temperature dependence of the widths is also included in this uncertainty budget.*
*The manuscript was changed accordingly (Page 11, line 13; Figs. 4, 5, 6, and 7):*
*"Line parameter uncertainties for water vapor and further trace gases were set according a combination of two uncertainty estimates: A first uncertainty specification is provided in the error codes of the aer_v3.2 line list provided alongside the LBLRTM radiative transfer model. The uncertainty of each parameter was assumed to correspond to the mean of the error range specified by the error code value. Since the error codes may not provide realistic uncertainty specifications for all spectral lines, an additional line parameter uncertainty estimate was obtained by taking the difference between the line parameters in the HITRAN 2008 database compared to the HITRAN 2012 database which was modified for FIR water lines according to the results of Delamere et al. (2010). To provide a conservative estimate, the uncertainty due to line parameter errors was set to the maximum value provided by these two alternative methods for each spectral point."*

**Page 13**
**5-9** – It is puzzling that the supposed dominant role of the H2O line parameters is mentioned first when, for the regions of interest in terms of continuum derivation, the dominant uncertainty is the H2O column (as shown in blue in Fig 4c). This paragraph should be reworded to emphasize the key conclusions of the uncertainty analysis as it pertains to the continuum.

*The wording was changed to emphasize the situation in the continuum retrieval windows (Page 13, line 5):" Figure 4d shows that the dominant contribution to the total uncertainty in the FIR is from IWV uncertainty, water vapor profile shape uncertainty and partly water vapor line parameters in the windows used for continuum retrieval, ..."*

**29** – It's unclear what the threshold of LWP < 100 has to do with snow accumulation on the LHATPRO.

*We thank for pointing out his unclear explanation. The following more precise outline was added in the revised manuscript to clarify the reason for the LWP threshold (Page 13, line 29): "As outlined above, clear-sky conditions are a prerequisite for the closure measurements. If, despite clear-sky conditions, the LHATPRO measurements indicate a high LWP, this indicates that snow has accumulated on the instrument and may bias the measurements. Therefore, we only selected spectra with LWP < 100 $g/m^2$."*

**Page 14**

**10** – Figure 4c indicates that the total uncertainty in the continuum channels between 400-500 cm$^{-1}$ range from 2-3 radiance units. However, in Figure 6b it doesn't look like the underlying gray uncertainty is that far from the red parts of the residual curve, especially from 470-500 cm$^{-1}$. Also, please define the residual to be obs-calc or calc-obs.

*Unfortunately, a preliminary version of Fig. 6 which was not consistent with the final uncertainty budget was mistakenly used in the initial manuscript. We thank the referee for pointing out this error.*
*Fig. 4 and 6 were updated to the final uncertainty budget. The figure caption was changed to incorporate the residual definition used in Fig. 6b: "(b) Mean spectral residuals (synthetic minus measured radiances)…"*

**pg 16** (major) The authors have chosen to obtain the H2O column by a retrieval in the FIR, which presents the possibility of circularity since the same instrument, uncertain line parameters,  etc. are being used in the column determination and the continuum derivation. Also, the continuum itself is an element of the column determination.
Clearly the authors must believe that this provides a better estimation of the column than alternative sources for the column. Similar closure studies (e.g. Turner et al., Delamere et al.) have derived the column from microwave measurements near lines that have line parameters with low uncertainty, removing potential circularity and lowering a key source of uncertainty. No details are given for the column retrieval by the LHATPRO, but it may use a similar approach as used in these other closure studies. It would be interesting for the authors to provide the rationale for their choice for determining the column. Why do they feel it provides a better value than the microwave?  How different are the column values obtained from each approach?  Is this difference a good estimate of the uncertainty in the column amount they are using?  A plot should be provided with these differences. What is the method used in the LHATPRO retrieval?

*The following text was added to the manuscript to provide additional information on the LHATPRO retrieval and to outline the reason for using the IWV fit described in Sect. A.1 (Page 16, line 11): "It measures sky brightness temperatures at 6 channels within the strong 183.31 GHz water vapor line with a repeat cycle of 1 s for IWV and 60 s for profiles (Radiometer Physics, 2013). The Radiometer Physics software (Radiometer Physics, 2014) allows for statistical retrieval of water vapor profiles which is based on a neuronal network approach (Jung et al., 1998) utilizing MMOD radiative calculations (Simmer, 1994) performed for a radiosonde training data set. However, the IWV results obtained with the LHATPRO show a significant bias compared to an IWV retrieval from solar FTIR spectra (Sussmann et al., 2009), which has been extensively validated against other instruments (see Sussmann et al., 2009; Vogelmann et al., 2011). The solar FTIR-based IWV retrieval is not suitable as an input to the FIR closure study is because few coincident measurements of AERI and solar FTIR are available. We therefore chose to implement the IWV retrieval procedure outlined below."*
*The difference between LHATPRO-based and retrieved IWV is discussed as follows (Page 17, line 4):"The mean correction relative to the LHATPRO first guess IWV was -0.098 mm, with a standard deviation of 0.089 mm. This corresponds to a mean IWV correction of 4.1% which is slightly beyond the mean fit uncertainty of 3.1%, i.e. the IWV fit leads to a significant improvement of the IWV input compared to using the LHATPRO data."*
*Figure A3 was adapted as suggested by the referee to show differences between fitted and LHATPRO IWV instead of ratios as in the initial manuscript.*

The definition and description of type-i and type-ii uncertainty (bottom of page) should be moved up to points ii or iii, instead of its current placement in point iv.

*The definition and description of type-i and type-ii uncertainty was moved up to point ii as suggested by the referee.*

(major) The uncertainty due to the line parameters is likely underestimated here. First, as above, the width errors should not be obtained from the error code in the line file.

Second, the temperature dependence of the widths need to be accounted for (assuming they are not). Lastly, comparing HITRAN 2012 to the aer line file that is used in this study shows that the width differences are not equally likely to be positive as negative – the signs of the differences are usually the same. Therefore, assuming that the resulting uncertainties are uncorrelated between different spectral regions is not appropriate. Although this type of correlation between different spectral regions may appear to be more accidental than other correlated errors (e.g. T profile), since the widths tend to come from calculations that are constrained to observed values, there may be a clear reason why they would generally be high (or low) in a particular version of the database. Reclassifying the line parameter errors as correlated (or somewhere in between correlated and uncorrelated) would increase the uncertainty in the column estimation.

*As outlined above, an alternative source of line parameter uncertainties (namely the difference between HITRAN 2008 and HITRAN 2012) was used as suggested by the referee. Uncertainty of the temperature dependence of line widths is included in the uncertainty analysis.*

*We agree with the referee on the fact that assuming no correlation between wavenumbers for line parameter errors may lead to an underestimation of the associated IWV uncertainty. Therefore, as suggested by the referee, the uncertainty was assumed to be partly correlated between wavenumbers. The manuscript was changed as follows to explain this (Page 16, line 19): "Line parameter errors may feature some correlation between wavenumbers due to systematic bias in the measurements used to constrain these parameters. To account for this, 50 % of the radiance uncertainty associated with line parameter errors for any spectral point was treated as correlated between wavenumbers (type ii), while the remaining 50 % were treated as uncorrelated (type i)."*

*Fig. A1-A3 and all corresponding results in the manuscript text were recalculated according to this modified uncertainty estimate. Note that further modifications to the IWV retrieval were made according to the suggestions of referee #1 (see first two comments of this reply).*

**19** – It is better to refer to this as "uncorrelated between wavenumbers" and "correlated between wavenumbers" rather than the current wording.

*The wording was changed as suggested by the referee.*

**Page17**
**11, 19** – "ensues" is not the correct word

*Page 17, line 11: "ensues" was changed to "can be calculated".*
*Page 17, line 19: "ensues" was changed to "we obtain"*

**pg 25 Table C1** contains key results and should be moved from an appendix into the main part of the paper

*The table was moved to the main part of the paper as suggested by the referee.*

[revised manuscript text omitted]

investigated the impact of switching from the CKD continuum model frequently used in climate models to a continuum model where absorption is enhanced at wavelengths greater than 1 μm based on recent measurements (CAVIAR). They found that for CKD and CAVIAR respectively, and relative to the no-continuum case, the solar component of the water vapor feedback is enhanced by about 4 and 9 %, the change in clear-sky downward surface irradiance is 7 and 18 % more negative, and the global-mean precipitation response decreases by 1 and 4 % (Rädel et al., 2015).

Due to the critical relevance of line parameter and continuum model uncertainties for climate simulations a series of quality measurement experiments has been performed. Such field closure studies comprise high-spectral-resolution radiance measurements and radiative transfer simulations of the measured spectra driven by coincident atmospheric state measurements of integrated water vapor (IWV) and other relevant parameters. As part of the U.S. Atmospheric Radiation Measurement (ARM) program (Ackermann and Stokes, 2003) a series of radiative closure experiments has been setup (e.g. Turner et al., 2004; 2012b) which was complemented by the Italian ECOWAR (Earth COoling by WAter vapor Radiation) project (e.g. Bhawar et al., 2008; Bianchini et al., 2011). Various experiments have addressed the quality of (water vapor) line parameters in the FIR (Esposito et al., 2007; Delamere et al., 2010; Masiello et al., 2012), the water vapor continuum in the FIR (Tobin et al., 1999; Serio et al., 2008; Delamere et al., 2010; Liuzzi et al., 2014), and the water vapor continuum in the MIR (Turner et al., 2004; Rowe et al., 2006; Rowe and Walden, 2009). A crucial requirement for radiative closure experiments in the FIR and MIR is to select a site guaranteeing a wide range of IWV levels including the occurrence of very low IWV levels. Dry atmospheric states (IWV < 1 mm) are highly beneficial to attain information on absorption coefficients in otherwise saturated spectral regions (e.g. the pure rational water band of water vapor). For these reasons, there have been dedicated campaigns performed in dry regions on the globe, e.g. at the Sheba ice station (Tobin et al., 1999) or the RHUBC I and RHUBC II campaigns carried out in Alaska and in the Atacama desert, respectively (Turner and Mlawer, 2010).

Coming to the NIR we note that for this spectral region to our knowledge no atmospheric radiative closure experiments have been reported in the literature with the exception of one study by Sierk et al. (2004). A hindrance for quantitative field studies may have been the fact that absorption in the NIR due to aerosols can become comparable to the magnitude of the water vapor continuum absorption of interest (Ptashnik et al., 2015). The possibility to accurately separate these two components depends on aerosol load (i.e. aerosol optical depth, AOD) and therefore on field site characteristics. We will come back to this when introducing the new Zugspitze field experiment (including NIR measurements) below. On the other side, there have been many laboratory studies in the NIR range. Laboratory experiments using FTIR spectrometry and large cells have shown that the self-continuum within bands contains more spectral structure than given by the MT_CKD model (Ptashnik et al., 2004; Paynter et al., 2009). The self-continuum absorption within the windows was found to be an order of magnitude stronger than given by MT_CKD and to vary from window to window significantly less. (Baranov and Lafferty, 2011; Ptashnik et al., 2011, 2013). Also the NIR foreign continuum in window regions was found to be about an order of magnitude stronger compared to MT_CKD (Ptashnik et al., 2012). Another issue is the obvious discrepancy between

laboratory measurements performed by different techniques. E.g. the magnitude of the self continuum in the 1.6 and 2.1 μm windows derived from laboratory FTIR spectrometry (Baranov and Lafferty, 2011; Ptashnik et al., 2011) is higher by about one order or magnitude as compared to results obtained by cavity ring-down spectroscopy (CRDS; Mondelain et al., 2013; 2015). Furthermore, CDRS results differ to laboratory results obtained by calorimetric interferometry (Bicknell et al., 2006) by a factor of 4-5. Reasons behind these inconsistent laboratory results could be the differing physical measurement principles and/or were tentatively attributed to surface effects (e.g. water on walls, water droplets or clusters in the cell; Ptashnik et al., 2013; 2015). Finally, a drawback common to all laboratory measurements is that they have to be performed at least at room temperature or even be heated, in order to detect the weak continuum absorption in the limited optical path length of the cells. Therefore, for climate and remote sensing applications an extrapolation of continuum coefficients to the lower atmospheric temperatures is required. This, however, is a problem because the observational evidence of a negative exponential temperature dependency of the self continuum still cannot be described by a physical model in a quantitative manner (e.g. Shine et al., 2012).

Coming to the NIR we note that for this spectral region to our knowledge no atmospheric radiative closure experiments have been reported in the literature with the exception of the studies by Sierk et al. (2004) and Mlawer et al. (2014). A hindrance for quantitative field studies may have been the fact that absorption in the NIR due to aerosols can become comparable to the magnitude of the water vapor continuum absorption of interest (Ptashnik et al., 2015). The possibility to accurately separate these two components depends on aerosol load (i.e. aerosol optical depth, AOD) and therefore on field site characteristics, as will be outlined when introducing the new Zugspitze field experiment below. On the other side, there have been many laboratory studies in the NIR range. Laboratory experiments using FTIR spectrometry and large cells have shown that the self- and foreign continuum within the windows was found to be significantly stronger than given by MT_CKD (Baranov and Lafferty, 2011; Ptashnik et al., 2011, 2012, 2013). Another issue is that laboratory measurements performed by different techniques have yielded to inconsistent results. For example, the magnitude of the self continuum in NIR windows derived from laboratory FTIR spectrometry is higher by about one order or magnitude compared to results obtained by cavity ring-down spectroscopy (CRDS; Mondelain et al., 2013; 2015), which furthermore differ significantly to laboratory results obtained by calorimetric interferometry (Bicknell et al., 2006). Finally, a drawback of laboratory measurements is that they are typically performed at least at room temperature or even heated, in order to detect the weak continuum absorption in the limited optical path length of the cells. Therefore, for climate and remote sensing applications an extrapolation of continuum coefficients to the lower atmospheric temperatures is required which may lead to significant inaccuracies due to the uncertainty of the self continuum temperature dependence (
[revised manuscript text omitted]

radiosonde measurements. We used radiosonde data from a campaign performed close to the Zugspitze site between Mar – Nov 2002 (for details see Sussmann and Camy-Peyret, 2002, 2003; Sussmann et al., 2009). The campaign data set comprises a number of 181 pairs of radiosondes launched with a 1-hour time separation, and each radiosonde pair has been combined to a best estimate of the state of the atmosphere according to the formalism by Tobin et al. (2006). Subsequently, both NCEP profiles and sonde-based Tobin-best-estimate profiles were normalized by IWV analogous to the analysis in the closure experiment described in Sect. 7, and then profile differences were computed. The red line in Fig.  4 shows the mean difference profile. The profile shape bias of 1.7 % given in Tab. 1 is just a simple proxy that has been obtained  as follows: for each pair of sonde and NCEP profiles, a difference vector was calculated. Each component of the  average  bias vector was then deduced as the mean of the absolute values of  the corresponding components of the difference vectors . The statistical profile shape uncertainty was set up via an error covariance matrix constructed from the difference profiles between NCEP and sonde-based Tobin-best-estimate profiles. This error covariance was used for the further statistical analysis of radiance uncertainty. Just to illustrate some properties of this covariance, the black error bars in Figure  4 show the 2-$\sigma$ statistical uncertainties of the difference profile (corresponding to the diagonal of the covariance). By calculating the mean of these error bars we can derive a simple scalar proxy for the statistical profile uncertainty of 9.4 % (Tab. 1). An estimate of the corresponding radiance uncertainty that includes the influence of layer-to-layer correlations can be obtained by multiplying the full error covariance matrix with the derivative matrix of radiance with respect to water vapor profile shape in the atmospheric layers (see Fig. 5) and its inverse. . This leads to the residual uncertainty shown in Fig. 8 (pink). The representation in Fig. 5 corresponds to the radiance change associated with a 1% change of water vapor density in a given altitude layer and subsequent rescaling of the profile to the IWV obtained as outlined in Sect. A.1. Due to the rescaling to a prescribed IWV, the 1%-increase of water vapor density in a given layer is associated with a decrease in all other layers. Therefore, a 1%-perturbation in the lowermost layer (2.96 - 4 km a.s.l.) corresponds to lowering the center of gravity of the water vapor profile and leads to a positive change in radiance, while for higher layers, the opposite is true. Due to the decrease of water vapor density with altitude (see Fig. 4b), the radiance effect of a 1%-perturbation decreases rapidly with altitude.

The temperature profiles used in the closure study are a composite of $T$ profiles retrieved from the ER-AERI spectra for the altitude range between the Zugspitze up to ~3.5 km a.s.l., while at higher altitude the $T$ profiles were set according to the NCEP reanalysis  as described in Sect. 4. The uncertainty estimate for these composite profiles was constructed from the same radiosonde campaign data as for the water vapor profile analysis outlined above. To generate an estimate of the uncertainty, synthetic radiance spectra were calculated using all radiosonde-derived best-estimate $T$ profiles from the campaign. The systematic part of the uncertainty was estimated by adding the ER-AERI calibration bias (0.66 %, see Tab. 1) and the estimated bias due to line parameter

uncertainties (see Sect. 6.2) to the synthetic radiance spectra. Then, the near-surface temperature profile retrieval described in Sect. 4 was applied to the modified radiances. Finally, the differences between our composite $T$ profiles and the radiosonde-based best-estimate profiles from the campaign were calculated (red line in Fig. 6B3). Note, that the sign of the bias below 3.5 km a.s.l. (see Fig. B36) is arbitrary in the sense that it depends on whether the calibration bias is added or subtracted. The random uncertainty of the composite $T$ profile was estimated by adding random error according to the statistical E-AERIER-AERI calibration uncertainty (0.13 %, Tab. 1) and E-AERIER-AERI noise (yellow line in Fig. 48) to the synthetic radiance spectra. Finally, the near-surface temperature profile retrieval described in Sect. 4 was applied to the modified radiances. This approach implicates that both the uncertainty due to the retrieval itself as well as additional uncertainty due to inaccurate radiance input are taken into account for the $T$ profile uncertainty estimate. An error covariance matrix estimate was then calculated from the difference of the radiosonde profiles to these composite $T$ profiles. Radiance uncertainties were then calculated by multiplication with the corresponding radiance derivative matrix depicted in Fig. B47. The resulting overall radiance uncertainties are shown in Figs. 4 and 5 (green).

[revised manuscript text omitted]

**Appendix A: Retrieval of IWV from ER-AERI spectra**

**A.1 Retrieval method**

We utilize an approach similar to the method proposed by Serio et al. (2008), i.e. IWV is retrieved via a derivative approach using one iteration to minimize ER-AERI vs. LBLRTM spectral residuals in IWV-sensitive windows. As first guess IWV, data from a LHATPRO microwave radiometer are used. LHATPRO (Radiometer Physics, Germany; Rose et al., 2005), designed for ultra-low humidity sites (IWV < 4.0 mm), is a microwave radiometer located side-by side to the ER-AERI. It measures sky brightness temperatures at 6 channels within the strong 183.31 GHz water vapor line with a repeat cycle of 1 s for IWV and 60 s for profiles (Radiometer Physics, 2013). The Radiometer Physics software (Radiometer Physics, 2014) allows for statistical retrieval of water vapor profiles which is based on a neuronal network approach (Jung et al., 1998) utilizing MMOD radiative calculations (Simmer, 1994) performed for a radiosonde training data set. However, the IWV results obtained with the LHATPRO show a significant bias compared to an IWV retrieval from solar FTIR spectra (Sussmann et al., 2009), which has been extensively validated against other instruments (see Sussmann et al., 2009; Vogelmann et al., 2011). The solar FTIR-based IWV retrieval is not suitable as an input to the FIR closure study is because few coincident measurements of AERI and solar FTIR are available. We therefore chose to implement the IWV retrieval procedure outlined below.

The procedure for selection of suitable spectral windows for IWV retrieval from the $400 - 600$ cm$^{-1}$ spectral range has been implemented as follows:

i) The uncertainty of the IWV fit for single spectral points is calculated for the remaining windows. IWV relative uncertainty is given as the residual uncertainty excluding IWV contribution divided by $\partial I/\partial$IWV, i.e. the derivative of downwelling spectral radiance $I$ with respect to IWV. The overall uncertainty comprises two classes of errors, namely type-i errors which are uncorrelated between wavenumber, and type-ii errors correlated between wavenumber. ER-AERI measurement noise is treated as a type-i error contribution (the underlying assumption being that line parameter errors for different lines are independent). Other uncertainty contributions such as ER-AERI calibration, T profile errors, and water vapor profile errors (see Sect. 6.3 for details) are correlated for different spectral channels (type ii). Line parameter errors may feature some correlation between wavenumbers due to systematic bias in the measurements used to constrain these parameters. To account for this, 50 % of the radiance uncertainty associated with line parameter errors for any spectral point was treated as correlated between wavenumbers (type ii), while the remaining 50 % were treated as uncorrelated (type i).

i) Spectral points (channels) are ordered from lowest to highest type-ii uncertainty.

iviii) Ensembles with stepwise increased number of channels are constructed including channels with increasing type-ii uncertainty, and the overall uncertainty (type i + ii) is calculated for each ensemble. Figure A1 shows this overall uncertainty depending on the number of included channels. E AERI measurement noise and line parameter errors are treated as type i error contributions (the underlying assumption being that line parameter errors for different lines are independent). Therefore, these Type-i contributions to the cumulative uncertainty are reduced by a factor $1/\mathrm{sqrt}(n)$ when $n$ channels are included in the fit (causing the decrease of uncertainty on the left hand side of Fig. A1). All other uncertainty contributions (E AERI calibration, T profile errors, and water vapor profile errors, see Sect. 6.3 for details) are correlated for different spectral channels, thereforeFor type-ii contributions, no uncertainty reduction is achieved by including more channels in the fit, and the overall uncertainty increases toward the right hand side of Fig. A1. This is because more and more channels with increasing type-ii uncertainty are included.

iv) The optimum number of spectral channels for the fit is deduced from the minimum of overall (type-i + ii) uncertainty (Fig. A1). The resulting optimum numbers of channels for the different spectra of our closure dataset are shown in Fig. A2; the mean value is 23.74.1 channels, with a minimum of 81 and a maximum of 410 channels.

v) The IWV fit according to steps i) - iv) is repeated for each iteration step of the continuum quantification procedure (see Sect. 7.3). This iterative approach serves to avoid interference between the continuum quantification and the IWV fit. Performing the IWV fit including only windows with negligible continuum contribution (i.e. excluding all windows with continuum uncertainty < 100 %) leads to a mean bias in the IWV results of 0.005 mm. This negligible bias indicates that the iterative approach is able to avoid significant interference between IWV fit and water vapor continuum determination.

The results of the IWV fit for all spectra included in the FIR closure data set are shown in Fig. A3. The mean correction relative to the LHATPRO first guess IWV was -0.09851 mm, with a standard deviation of 0.07589 mm. This corresponds to a mean IWV correction of 4.1%, which is slightly beyond the mean fit uncertainty of 3.1%, i.e. the IWV fit leads to a significant improvement of the IWV input compared to using the LHATPRO data.

**A.2 Uncertainty estimate**

[revised manuscript text omitted]

**Figures**

[revised manuscript text omitted]

5 **Figure A3.** Adjustment $\Delta$IWV _to the first guess value_ IWV$_{LHATPRO}$ derived in the IWV fit for the spectra included in the FIR continuum data set.